# Water orientation and dynamics in the closed and open influenza B virus M2 proton channels

Martin D. Gelenter [1,2], Venkata S. Mandala [1,2], Michiel J. M. Niesen [1,2], Dina A. Sharon[1,2], Aurelio J. Dregni[1], Adam P. Willard[1] & Mei Hong [1✉]

The influenza B M2 protein forms a water-filled tetrameric channel to conduct protons across the lipid membrane. To understand how channel water mediates proton transport, we have investigated the water orientation and dynamics using solid-state NMR spectroscopy and molecular dynamics (MD) simulations. [13]C-detected water [1]H NMR relaxation times indicate that water has faster rotational motion in the low-pH open channel than in the high-pH closed channel. Despite this faster dynamics, the open-channel water shows higher orientational order, as manifested by larger motionally-averaged [1]H chemical shift anisotropies. MD simulations indicate that this order is induced by the cationic proton-selective histidine at low pH. Furthermore, the water network has fewer hydrogen-bonding bottlenecks in the open state than in the closed state. Thus, faster dynamics and higher orientational order of water molecules in the open channel establish the water network structure that is necessary for proton hopping.

[1] Department of Chemistry, Massachusetts Institute of Technology, Cambridge, MA, USA. [2] These authors contributed equally: Martin D. Gelenter, Venkata S. Mandala, Michiel J. M. Niesen, Dina A. Sharon. ✉email: meihong@mit.edu

Water is ubiquitous in the interior of membrane proteins and co-regulates protein function. Substrate and ion transport by membrane proteins requires internal water molecules with the appropriate dynamics[1–3]. When water itself is transported, water–water hydrogen bonding dictates the rate of transport[4]. In contrast to most substrates and ions, protons are transported by water molecules as charge defects rather than atoms, by means of proton exchange and rapid rearrangement of water–water hydrogen bonds[5]. Transport of charge defects via this Grotthuss mechanism requires reorientation of water molecules and hydrogen bonds along the conduction pathway[6,7]. When the hydrogen bonding network is broken, for example, by polar amino acid residues in a protein, then proton transport can be prevented while water transport can remain, as in aquaporin[8,9]. When a protein channel contains more than a single file of water molecules and when proton conduction is predominantly unidirectional, then both water dynamics and water–water hydrogen bonding are expected to affect proton transport[10]. The interplay among water dynamics, water–water hydrogen bonding, and water orientation for proton transport is not yet well understood. This is partly due to the experimental challenges of determining atomic-resolution structures of membrane protein channels. Even when cryogenic-temperature crystal structures become available, usually only static snapshots of thermodynamically favored water are detected[9,11,12]. These snapshots reveal the oxygen positions of the structured water but not the disordered water, and the O–H bond orientation is usually unknown. In the absence of experimental data, molecular dynamics simulations have provided the main source of information about water dynamics and water orientation that underlie proton transfer[6,7,13].

The M2 protein of influenza viruses forms a water-filled proton channel[14] that serves as a model for understanding how proton transport is mediated by the dynamic structures of water and protein. M2's proton transport pathway is lined by four subunits of the transmembrane (TM) protein[15]. The channel is activated at low pH, when a central histidine residue becomes protonated[16]. Under this acidic condition, the histidine imidazolium rings, which are hydrogen-bonded to water, reorient on the microsecond timescale to shuttle protons from N-terminal water molecules to C-terminal water, causing proton flux into the virion[17,18]. A tryptophan (Trp) residue that is one helical turn away blocks the C-terminal protons from protonating the histidine, thus only allowing proton conduction from the N-terminus to the C-terminus[19–21]. Therefore, M2 transports protons using a mixed hydrogen-bonded chain between water and histidine. This detailed information about the histidine structure and dynamics in the influenza A M2 (AM2) protein was obtained from extensive solid-state NMR and crystallographic data[17,18,22–24]. In comparison, several aspects of the water structure and dynamics in the M2 pore have not been addressed. First, the water orientations in the closed and open channels are not known for M2, nor for most membrane proteins. Are water molecules partially oriented to permit Grotthuss hopping, and if so does the water orientation differ between the closed and open states? Second, it is not known how residues other than histidine affect the water–water hydrogen-bonding network. While AM2's pore-lining residues are nonpolar, the influenza B virus M2 protein (BM2) contains three serine residues along its pore-lining surface (Fig. 1). This polar surface might be expected to affect protein–water interactions and water–water hydrogen bonding, in turn changing the proton transport behavior. Third, for BM2, how water dynamics differ between the closed state and open state is not known.

To answer these questions, we have now conducted solid-state NMR experiments and MD simulations on membrane-bound

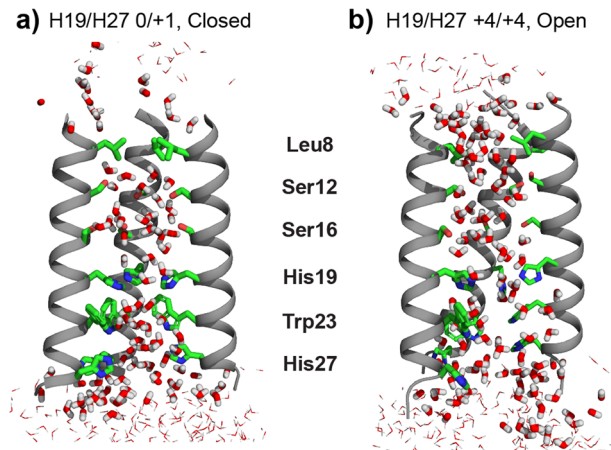

**Fig. 1 Equilibrated BM2 structures in the POPE membrane. a** Closed channel with the His19 and His27 tetrads in the 0 and +1 charge states, respectively, to mimic their protonation states at high pH. **b** Open channel with both His19 and His27 tetrads in the +4 charge state to mimic their protonation states at low pH. For clarity, only three out of four helices are shown. Pore-lining residues Leu8, Ser12, Ser16, His19, Trp23, and His27 are shown as sticks. Water in the channel lumen is shown as sticks while water outside the channel is shown as lines.

BM2. Solid-state NMR is an attractive method for obtaining experimental information about the dynamics and orientation of water molecules under biologically relevant conditions[25,26]. A direct comparison of water properties between the open and closed BM2 channels is possible due to our recently determined high-resolution structures of BM2 in lipid bilayers, where the 10 lowest-energy structures have a heavy-atom root mean-squared deviation of 1.5 Å[27]. These structures show that the four TM helices increase the tilt angle by 6° at low pH than at high pH, and the helices are on average 2.0 Å more separated from each other (Fig. 1). This uniform expansion of the BM2 pore at low pH correlates with the ability of BM2 to conduct protons bidirectionally, down the concentration gradient, like a canonical ion channel. In comparison, AM2's TM domain exhibits a helical kink at G34, which acts as a hinge to alternate water access to the N- or C-terminal halves of the pore[11,24]. This distinct conformational motion correlates with AM2's function to conduct protons strictly inward, like a transporter. On the basis of the open and closed BM2 structures, we now investigate the dynamics and orientations of water molecules in the BM2 pore. We conduct solid-state NMR experiments and MD simulations at 273–277 K, where both disordered and structured water molecules are present. We show that BM2's pore is several water molecules wide with many possible hydrogen-bonding pathways. Solid-state NMR data show that water in the low-pH BM2 channel exhibits faster reorientations as well as higher orientational order compared to the high-pH channel. This unexpected result suggests that water undergoes rapid small-amplitude reorientations to mediate proton conduction. MD simulations give consistent findings about the relationship between water orientation and dynamics, and indicate that this relationship is controlled by the proton-selective histidine residue.

## Results

BM2 conducts protons at low pH, when the proton-selective histidine (H19) tetrad becomes protonated. A second, membrane-surface histidine (H27) is also protonated at low pH, but is not essential for proton conduction[28,29]. Experimentally, the closed and open states are accessible by controlling the pH of the membrane samples to pH 7.5 ("high pH") and pH 4.5 ("low pH"),

respectively. These conditions lead to average protonation states of 0/+1 for the H19/H27 tetrads at pH 7.5 and +4/+4 at pH 4.5, based on measured $pK_a$ values[29,30]. For all-atom MD simulations of the solvated BM2 channel, we represent the closed and open channels by charge states of 0/+1 (closed[0/+1]) and +4/+4 (open[+4/+4]) for H19/H27, respectively, while the pH was not directly controlled. Simulations were initialized with the energy-minimized 1.5 Å experimental structures of BM2 tetramers in a POPE bilayer, matching experimental conditions (see Methods section; Supplementary Fig. 1a, b). Both charge states (Fig. 1) were stable, with Cα root mean-squared deviations (RMSDs) of 2.3–2.4 Å (Supplementary Fig. 1c, d). Channel-water dynamics were analyzed from four independent 100 ns MD trajectories for each channel. Below, we present findings from solid-state NMR experiments and MD simulations in parallel to facilitate comparison.

**The BM2 channel pore is more hydrated at low pH than at high pH.** We measured the amount of water in the high-pH and low-pH BM2 channels using [13]C-detected water [1]H polarization transfer experiments (Fig. 2a and Supplementary Fig. 2a). The water [1]H magnetization was selected by a soft [1]H excitation pulse followed by a 0.286-ms [1]H $T_2'$ filter, then transferred to protein protons during a mixing time, $\tau_{mix}$. The transferred magnetization was detected through protein [13]C signals[25,26]. All aliphatic [13]C intensities were integrated for the buildup curve analysis. The intensity ratio between spectra measured at short and long mixing times is inversely proportional to the protein–water distance (Supplementary Method, Eq. 7), whereas the relative intensities of the closed and open channels in the short-$\tau_{mix}$ limit reflect the relative magnitude of the product of the water amount and the square root of the effective spin diffusion coefficient (Supplementary Method, Eq. 5). This effective spin diffusion coefficient depends on both the chemical exchange rate and the [1]H spin diffusion coefficient[26,31]. It is known that chemical exchange between water and labile protein protons is slower at low pH than at high pH. For example, labile imidazole protons have hydrogen exchange rates of ~100,000 $s^{-1}$ at pH 7.5 but only ~10,000 $s^{-1}$ at pH 4.5[18,32], and the indole $H^N$ proton of Trp has a hydrogen

exchange rate of ~10 $s^{-1}$ at pH 7.5 but only ~0.1 $s^{-1}$ at pH 4.5 (ref. [33]). Serine hydroxyls have similar hydrogen exchange rates at pH 4.5 and pH 7.5, but both are slow at ~300 $s^{-1}$ (ref. [34]). Finally, exchange of protected amide protons with water is expected to be too slow to affect magnetization transfer on the millisecond timescale of $\tau_{mix}$ used here[35]. Despite the slower chemical exchange rates at low pH, the pH 4.5 BM2 sample shows faster polarization transfer buildup rates than the pH 7.5 sample (Fig. 2a, b and Supplementary Data 1), indicating that the low-pH channel pore contains more water[36,37]. The intensity ratios between the low-pH and high-pH samples extrapolate to 1.95 at vanishing $\tau_{mix}$ (Fig. 2c), indicating that the low-pH channel contains at least 1.95 times as much water as the high-pH channel.

MD simulations yielded both average and site-specific information about the number of water molecules in the open and closed states. We found that the open[+4/+4] BM2 channel contains 120 ± 7 water molecules from a channel-axis coordinate of −16 Å to +16 Å, while the closed[0/+1] channel contains 60 ± 5 water molecules (Fig. 2d and Supplementary Movie 1). Thus, the number of water molecules is 2.0 times higher in the low-pH channel than the high-pH channel (Fig. 2e), in excellent agreement with the experimentally measured ratio of water amounts. The number of water molecules increases the most near the hydrophobic valve formed by L8, and in the H19-W23 proton conduction motif. This increase in water density coincides with an increase in the channel diameter near those residues (Supplementary Fig. 1b).

**Water motions are faster in the open channel than the closed channel.** To investigate water dynamics in the low-pH and high-pH BM2 channels, we measured water protons' apparent transverse relaxation times ($T_2'$) and rotating-frame relaxation rates ($R_{1\rho}$). The $T_2'$ values are sensitive to picosecond (ps) to nanosecond (ns) motions while the $R_{1\rho}$ values are sensitive to microsecond (µs) to millisecond (ms) motions. These relaxation rates were measured at a sample temperature of 273 K to minimize chemical exchange between water and labile protein protons. Based on the composition of our membrane samples, about two

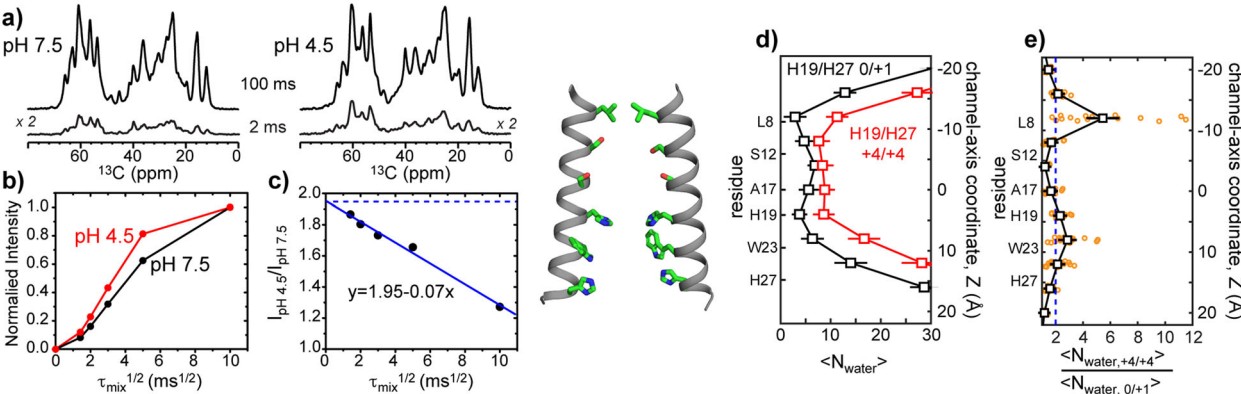

**Fig. 2 Experimental and simulated water amounts in closed and open BM2 channels. a** Water-transferred [13]C spectra with 2 ms and 100 ms [1]H mixing times at pH 7.5 and pH 4.5. Note the higher intensities of the 2-ms spectrum of the low-pH sample compared to the high-pH sample. **b** Normalized water-transferred intensity ($S/S_0$) as a function of mixing time $\tau_{mix}$. The protein shows faster intensity buildup at low pH than at high pH. **c** Ratio of the low-pH versus high-pH water-transferred intensities as a function of [1]H mixing time. The ratio has an intercept of 1.95 (blue dashed line), indicating that the low-pH channel contains at least 1.95-fold more water than the high-pH channel. **d** Simulated average number of water molecules in 4 Å bins along the channel axis in the open (red) and closed (black) states. A ribbon view of the BM2 TM domain is shown on the left to illustrate the positions of key residues (sticks). For clarity, only two out of four TM helices are shown. **e** Ratio of the simulated number of water molecules for the open and closed states in 4 Å bins along the channel axis. On average, 2.0 times more water molecules are found in the open channel (blue dashed line), consistent with the experimental data. Black squares and error bars for simulations correspond to the mean and standard error of the mean for four independent trajectories. The data from the individual trajectories are shown as open orange circles.

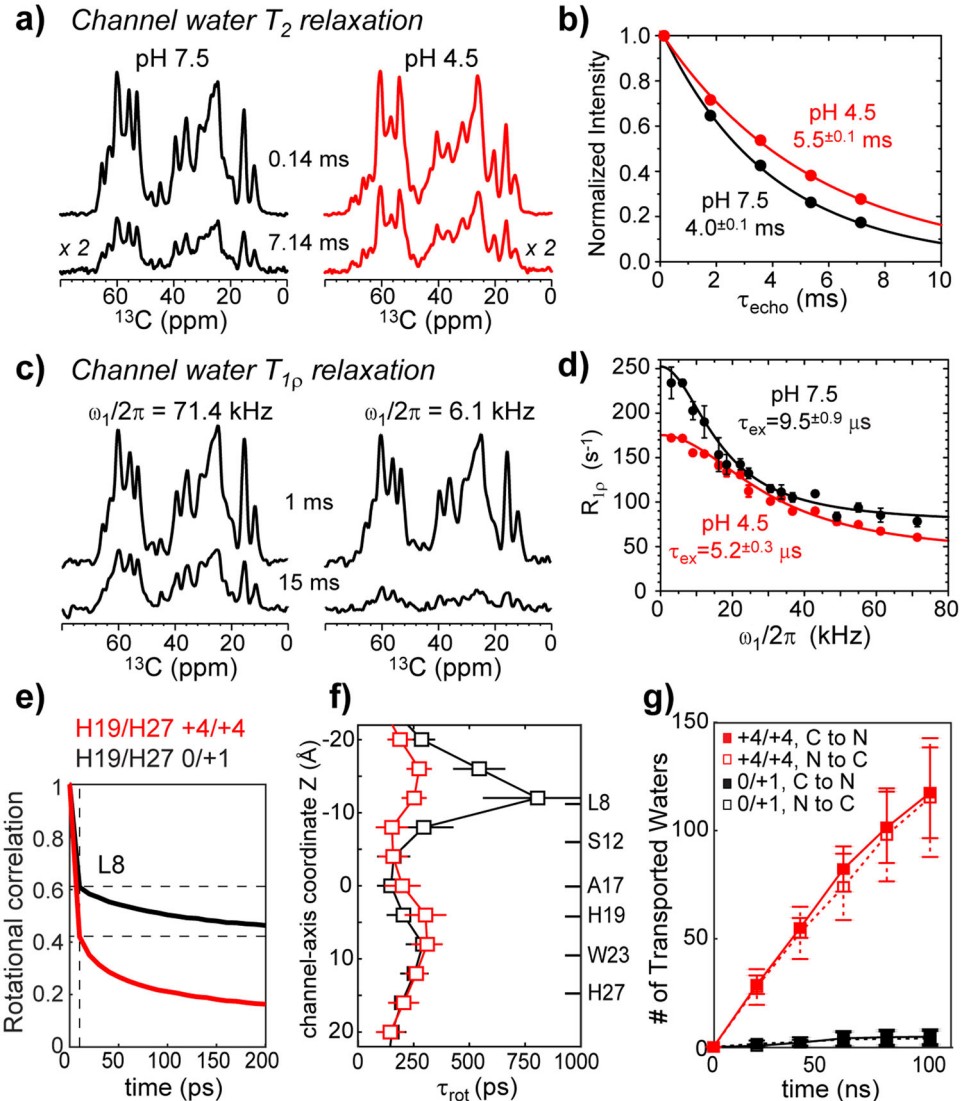

**Fig. 3 Rotational and translational diffusion of water in BM2 channels. a** Representative $^{13}$C-detected NMR spectra probing water $^1$H $T_2'$ at pH 7.5 and pH 4.5. These spectra were measured with total echo delays of 0.14 ms and 7.14 ms. **b** Channel-water $^1$H $T_2'$ relaxation decay curves. $T_2'$ relaxation is slower at low pH (red) than at high pH (black), indicating that the low-pH water has faster rotational dynamics. **c** Representative $^{13}$C-detected NMR spectra of high-pH BM2 probing channel-water $R_{1\rho}$ relaxation. The spectra were measured with spin-lock times of 1 ms and 15 ms and spin-lock field strengths of 71.4 kHz (left) and 6.1 kHz (right). **d** On-resonance $^1$H $R_{1\rho}$ relaxation dispersion profiles of water in the pH 7.5 (black) and pH 4.5 (red) channel. The low-pH channel shows a shorter $\tau_{ex}$ than the high-pH channel. Error bars for $R_{1\rho}$ values are the standard deviation of the fit for the signal decay at each spin-lock strength. **e** Simulated rotational correlation decay curves of waer near L8 in the closed and open channels. A fast decay in ~10 ps followed by a slower decay in ~500 ps is seen, and is representative of all water near protein residues. **f** Slow-component water rotational correlation times along the channel axis for closed and open BM2 channels. In the open state, water near H19 has the longest $\tau_{rot}$. In the closed state, water near L8 has the longest $\tau_{rot}$. **g** Total number of water molecules transported across the channel in both directions for the high-pH and low-pH states. Error bars for simulations are the standard error of the mean for four independent trajectories.

thirds of all water is bulk-like, one third is associated with the membrane surface, and only ~2% is within the channel[25]. We define bulk-like water as highly dynamic interlamellar water that does not exchange with the lipid headgroup or protein protons, but that is still not as mobile as isotropic bulk water. We used $^1$H-detected NMR experiments to probe the dynamics of the bulk-like and lipid-associated water (Supplementary Fig. 3a, b) and $^{13}$C-detected experiments to probe the dynamics of channel water near protein residues (Supplementary Fig. 2a, b).

At low pH, the $^1$H-detected water $T_2'$s are 48 ms for bulk-like water and 2.2 ms for lipid-associated water (Supplementary Fig. 3e and Supplementary Table 1). The fast relaxation of the lipid-associated water is consistent with retardation of water dynamics

by the membrane surface and chemical exchange of water protons with labile lipid headgroup protons. In comparison, $^{13}$C-detected channel water displays an intermediate $T_2'$ of 5.5 ± 0.1 ms at low pH (Fig. 3a, b), and the value decreases to 4.0 ± 0.1 ms at high pH. Transverse relaxation is driven by molecular tumbling[38], with slower relaxation indicative of shorter rotational correlation times ($\tau_{rot}$). Because the channel-bound water has a small degree of orientational anisotropy (vide infra), coherent effects can accelerate the apparent transverse relaxation compared to that caused by molecular tumbling alone. Despite this coherent contribution, we can estimate an upper limit of $\tau_{rot}$ using the Bloembergen-Purcell-Pound (BPP) theory (Eq. 2, Methods section)[38]. The upper limit $\tau_{rot}$ values that correspond to these $^{13}$C-detected water

[1]H $T_2'$'s are 5.5 ns for the low-pH water and 7.6 ns for the high-pH channel water. Both correlation times are an order of magnitude longer than the [1]H-detected bulk-like water $\tau_{rot}$ of 0.5–0.7 ns (Supplementary Table 1). Thus, the channel-water dynamics is highly restricted compared to bulk-like water. But even the bulk-like water in these hydrated liposomes is much less mobile than true bulk water, which has been reported to have a correlation time of 5.8 ps at 275 K[39]. We searched for residue-specific differences in the channel-water dynamics. However, even at a very short [1]H mixing time of 0.1 ms, all [13]C-detected [1]H $T_2'$ values are similar, indicating that water translational diffusion and [1]H spin diffusion are rapid on this timescale, thus obscuring site-specific differences. Therefore, for the [13]C-detected water [1]H $T_2'$ relaxation measurements, we used a 4-ms [1]H mixing time and integrated the aliphatic [13]C intensities to increase the sensitivity of the measurement.

To investigate slower dynamics of the channel water, we measured water [1]H $R_{1\rho}$ dispersion with protein [13]C detection, which distinguishes the protein-proximal channel water from bulk-like water and lipid-associated water (Fig. 3c and Supplementary Fig. 2b)[40]. If two water populations with different isotropic chemical shifts exchange on the µs to ms timescale, then $R_{1\rho}$ relaxation rates will vary with the spin-lock field strength in the kilohertz range. Relaxation is faster under weaker spin-lock field strengths and slower under stronger spin-lock fields. The intrinsic rotating-frame spin-lattice relaxation rate ($R_{1\rho}^0$) is obtained in the limit of infinitely strong spin-lock fields[41]. Figure 3d shows that both the high-pH and low-pH channel water displays significant $R_{1\rho}$ dispersion in a spin-lock field strength range of 3–71 kHz. Fitting the profile using a two-state model[42] (Supplementary Method) yielded exchange time constants ($\tau_{ex}$) of 5.2 ± 0.3 µs for the low-pH sample and 9.5 ± 0.9 µs for the high-pH sample. The fitting also yielded an exchange amplitude ($\phi_{ex}$) of 2.6 ± 0.2 × 10[7] rad[2]s[−2] for the low-pH sample and 1.9 ± 0.2 × 10[7] rad[2]s[−2] for the high-pH sample. The exchange amplitude depends on the populations of the two states ($p_1$ and $p_2$) and chemical shift difference ($\Delta\omega$) according to $\phi_{ex} = p_1 p_2 \Delta\omega^2$. Assuming comparable populations, these $\tau_{ex}$ values indicate a [1]H chemical shift differences of ~2 ppm between the two populations of water. Based on considerations of the chemical exchange rates, we attribute the exchange process to that between two pools of water molecules in the channel pore (Supplementary Method)[43,44]. Importantly, this exchange process is 2-fold faster in the low-pH channel than in the high-pH channel. In comparison, bulk-like water shows no $R_{1\rho}$ relaxation dispersion while the lipid-associated water exhibits only minimal dispersion, which is independent of pH (Supplementary Fig. 3c, d, f). Finally, the intrinsic relaxation rate $R_{1\rho}^0$ is 2–10-fold faster for the channel water than the bulk-like and lipid-associated water (Supplementary Table 2), indicating that channel-water motion is more restricted than bulk-like and lipid-associated water[45,46].

MD simulations of water rotational dynamics are fully consistent with these experimental results. In both the open[+4/+4] and closed[0/+1] channel, the rotational correlation of water decays over two distinct timescales (Fig. 3e). We attribute the fast decay within ~10 ps to thermal motion of bulk-like water, and the slow decay to motion of the hydrogen-bonded water molecules in the confined pore and interactions of water with protein residues. Due to the time resolution used in the analysis of the MD simulations, we cannot further comment on the sub-10-ps rotational time, although it is possible that this motion is also affected by confinement within the channel[47,48]. The long-time relaxation component is slower for the closed[0/+1] state than the open[+4/+4] state, with an average $\tau_{rot}$ of 253 ± 12 ps for the closed channel and 223 ± 6 ps for the open[+4/+4]

channel. Moreover, the slow-component's $\tau_{rot}$ values are not uniform throughout the channel (Fig. 3f): the slowest rotational dynamics is observed for water near L8 in the closed[0/+1] state and for water near H19 in the open[+4/+4] state.

We also examined water flux through the channel, by counting the number of water molecules that entered the channel on one side and exited at the other side (counted as one transport event). The open[+4/+4] channel transports 1.19 ± 0.24 water molecules per nanosecond (averaged over both directions) while the closed[0/+1] channel transports only 0.04 ± 0.03 water molecules per nanosecond (Fig. 3g and Supplementary Movie 1). The 30-fold reduction of the number of transported water molecules by the closed channel is striking. In addition, a water molecule takes 7.6 ± 0.2 ns on average to pass through the open[+4/+4] channel and 26 ± 3.4 ns to pass through the closed[0/+1] channel. The lifetime of an excess proton on hydronium ions is only a few ps[49]. Therefore, water translational diffusion through the channel is too slow to be the dominant process for proton transport, supporting the H19-mediated proton shuttling mechanism. We next computed the mean-squared displacement (MSD) of water molecules in the channel (Supplementary Fig. 4b), and fit it to the equation $MSD(t) = 6Dt^\alpha$, where the prefactor $D$ corresponds to the water translational diffusion coefficient when the exponent $\alpha = 1$. We find that the water translational motion is sub-diffusive in both channels ($\alpha < 1$), but the dynamics are even slower in the closed[0/+1] state than in the open[+4/+4] state (Supplementary Fig. 4c). The differences in the water dynamics between the open[+4/+4] state and closed[0/+1] state are illustrated in Fig. 4 and Supplementary Fig. 5.

**Water is more oriented in the open BM2 channel than in the closed channel.** Water rotational diffusion can facilitate the making and breaking of hydrogen bonds. If the hydrogen-bonded network differs between the closed and open channels, then we would expect the water orientations to differ as well. In general, water in hydrated membranes is expected to be predominantly isotropic on NMR timescales. Even if there is anisotropy for the 2% of water that resides in the channel, this anisotropy is expected to be small and hence difficult to measure accurately by conventional lineshape experiments[50]. Therefore, we probed channel-water orientation using [1]H spin-lock recoupling experiments[51] (Supplementary Fig. 2c and Supplementary Fig. 3c). Using spin-lock rf field strengths ($\omega_1$) that fulfill three recoupling conditions, $\omega_1 = 0.5\omega_r$, $\omega_r$, and $2\omega_r$, where $\omega_r$ is the MAS frequency[52], we selectively recouple nuclear spin interactions. For water, at the $\omega_1 = \omega_r$ and $2\omega_r$ conditions, [1]H chemical shift anisotropy (CSA) is recoupled, while at the $\omega_1 = 0.5\omega_r$ and $\omega_r$ conditions, [1]H–[1]H dipolar couplings are recoupled. The recoupled interactions should cause coherent oscillations of the time-dependent intensity decays, which can be fit to extract the motionally averaged anisotropies. The rigid-limit water [1]H CSA is 28.5 ppm[53], which corresponds to 22.8 kHz at 800 MHz, while the rigid-limit [1]H–[1]H dipolar coupling within each water molecule is 35 kHz. As before, [13]C detection allows us to selectively detect water molecules inside the channel, near protein residues.

Figure 5a shows the [13]C-detected channel-water [1]H intensity decays under spin-lock recoupling. For both low- and high-pH samples, we observed clear oscillations at $\omega_1 = \omega_r$ and $2\omega_r$ but not at $\omega_1 = 0.5\omega_r$, indicating that the channel water has finite motionally averaged [1]H CSA but vanishing [1]H–[1]H dipolar couplings. The latter can be understood by proton exchange between water molecules, which averages the intramolecular [1]H–[1]H dipolar coupling. To verify that the oscillations at $\omega_1 = \omega_r$ and $2\omega_r$ are indeed due to [1]H CSA, we conducted the REfocused

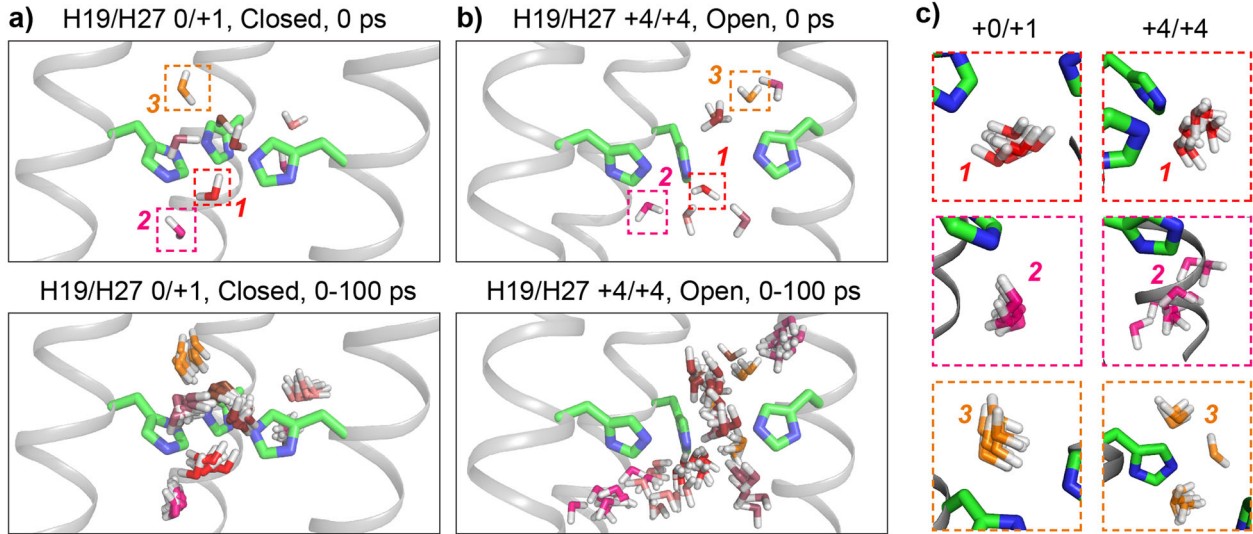

**Fig. 4 Representative snapshots of rotational and translational diffusion of water molecules within 3Å of H19. a** Water molecules near H19 in the closed$^{0/+1}$ channel. **b** Water molecules near H19 in the open$^{+4/+4}$ channel. The composite views at the bottom are taken in 10 ps steps over a 100-ps interval. For clarity, the oxygens of individual water molecules are colored differently. **c** Zoomed-in views of the trajectories of individual water molecules highlighted in **a** and **b**.

CSA Rotating-frame Relaxation (RECRR) experiment, which refocuses the CSA using phase-inverted spin-lock and 180° pulses (Supplementary Fig. 2d)[54]. Indeed, RECRR abolished most of the oscillations at $\omega_1 = \omega_r$ and $2\omega_r$ while retaining the same smooth decay at $\omega_1 = 0.5\omega_r$ (Fig. 5b), confirming that $^1$H CSA is the main mechanism of the coherent oscillation.

To compare the channel-water anisotropy with the anisotropy of lipid protons and other water protons, we conducted the spin-lock recoupling experiments with $^1$H detection (Supplementary Fig. 3). The lipid $CH_2$ and $CH_3$ protons exhibit coherent oscillations at $\omega_1 = 0.5\omega_r$ and $\omega_r$ but negligible oscillation at $\omega_1 = 2\omega_r$ (Supplementary Fig. 3g). This is consistent with the fact that lipid chains are anisotropically mobile and each $CH_2$ group has significant geminal $^1$H–$^1$H dipolar couplings while the $^1$H CSA is weak[55,56]. The total water $^1$H signal in the $^1$H spectra shows coherent oscillations at $\omega_1 = \omega_r$ and $2\omega_r$ but not at $\omega_1 = 0.5\omega_r$ (Supplementary Fig. 3f), similar to the $^{13}$C-detected channel-water behavior, indicating CSA recoupling. However, the minima of the total water oscillations are much higher than the minima of $^{13}$C-detected channel-bound water (Fig. 5a), indicating that only a fraction of all water is anisotropic. We attribute this anisotropy to water interacting with the membrane surface.

Importantly, the $^{13}$C-detected $^1$H spin-lock recoupling data indicate that the CSA-induced oscillation is faster at low pH than at high pH (Fig. 5a), indicating that the water orientational order is larger in the low-pH channel than in the high-pH channel. Although $^1$H CSA is the dominant coherent effect for water, contributions from $^1$H–$^1$H homonuclear dipolar couplings can speed up the dephasing at the $\omega_r$ recoupling condition[51]; therefore, we report CSA parameters extracted from the $2\omega_r$ matching conditions to compare the water CSAs between the high- and low-pH samples. Fitting the oscillations by numerical simulations (Supplementary Fig. 6) yielded a motionally averaged $^1$H CSA of $1.8 \pm 0.2$ ppm at low pH and $1.2 \pm 0.2$ ppm at high pH for the $\omega_1 = 2\omega_r$ condition. These values correspond to order parameters $S_{CSA}$ of $0.063 \pm 0.007$ at low pH and $0.042 \pm 0.007$ at high pH[53].

We next compared the simulated water orientational order with the experimental values by analyzing the O–H bond order parameter, $S_{OH}$, of water molecules in the MD simulations.

The principal axis of the water $^1$H CSA is roughly colinear with the O–H bond, thus $S_{CSA}$ is equivalent to $S_{OH}$. The average water $S_{OH}$ value in the open$^{+4/+4}$ state is $0.039 \pm 0.005$, which is slightly larger than the average water $S_{OH}$ value of $0.029 \pm 0.004$ in the closed$^{0/+1}$ state (Fig. 5c). Both values are in remarkably good agreement with the experimentally measured water $S_{CSA}$ at high and low pH. The simulated $S_{OH}$ values are dominated by water near a few residues, especially L8 and H19. To better understand these results, we extended our analysis by quantifying the water orientation in terms of the angle between the vector along the O–H bond and the channel axis (Fig. 5d). The site-resolved orientational averages (Figs. 5e and 6a and Supplementary Fig. 7) show that water in the closed$^{0/+1}$ channel displays little orientational preference except near L8, where there is a moderate preference for the oxygen-down orientation. In contrast, in the open$^{+4/+4}$ state, water orientational order is seen at almost all positions along the channel. Notably, this open$^{+4/+4}$ state orientational polarization is negative (oxygen pointing to the C-terminus) in the N-terminal half of the channel while positive (oxygen pointing to the N-terminus) in the C-terminal half of the channel (Figs. 5e and 6a and Supplementary Movie 1). This orientational switch occurs at H19. In other words, water molecules in the channel preferentially align with their oxygens pointing toward the charged H19 (Fig. 6c).

The closed$^{0/+1}$ and open$^{+4/+4}$ states differ in terms of the BM2 backbone conformation as well as the charge state of H19 and H27. To evaluate which of these factors cause the water anisotropy in the open$^{+4/+4}$ state, we conducted simulations in which the protein conformation was restrained to the low-pH solid-state NMR structure while the H19 and H27 charge states were assigned as $0/+1$. Conversely, we restrained the protein conformation to the closed high-pH solid-state NMR structure while enforcing the $+4/+4$ charge state. These simulations show that the water anisotropy was abolished in the artificial open$^{0/+1}$ state despite the open protein conformation, while the orientational trend of oxygen-up C-terminal to H19 and oxygen-down N-terminal to L8 is maintained in the closed$^{+4/+4}$ state, despite the closed protein conformation (Fig. 5f). Therefore, the water orientational order is primarily controlled by the charge state of H19 and H27 tetrads, not the distances between helices.

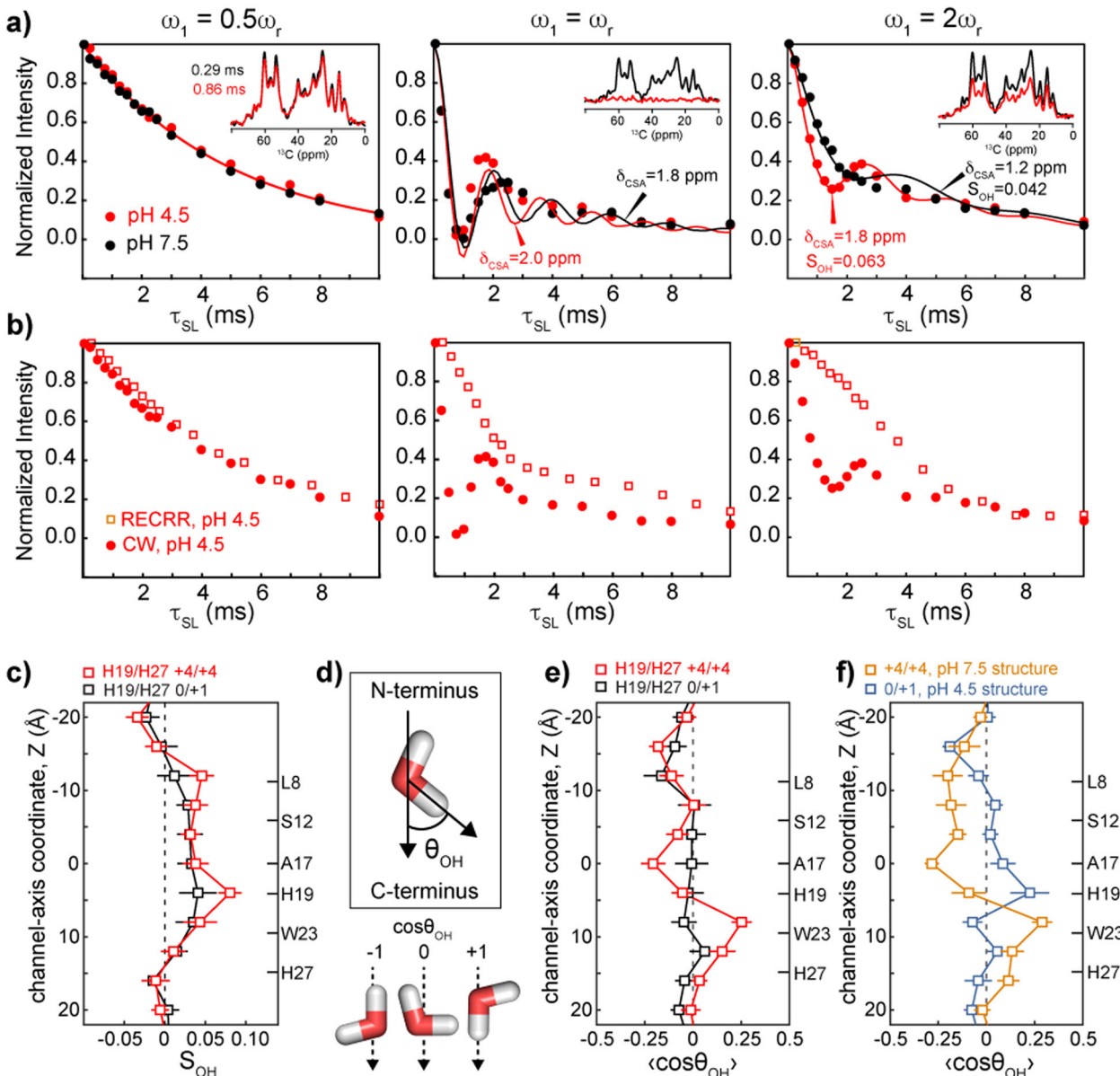

**Fig. 5 Channel-water orientational order in BM2 from $^{13}$C-detected $^1$H spin-lock recoupling NMR experiments and MD simulations. a** $^{13}$C-detected water $^1$H spin-lock recoupling data at $\omega_1 = 0.5\omega_r$ (left), $\omega_r$ (middle) and $2\omega_r$ (right). The spectral region integrated for each data point is shown in the inset for representative spin-lock times. Coherent oscillations are observed at $\omega_r$ and $2\omega_r$ but not at $0.5\omega_r$, indicating that $^1$H CSA is the dominant anisotropic interaction. **b** Comparison of $^{13}$C-detected water $^1$H RECRR data (open squares) and CW spin-lock recoupling data (filled circles) at the three recoupling conditions. The oscillations are mostly removed by RECRR. **c** Simulated water OH-bond order parameter, $S_{OH}$, in the open and closed BM2 channels. **d** Definition of $\theta_{OH}$ and representative water orientations for $\cos\theta_{OH}$ values of $-1$, 0, and 1. Positive $\cos\theta_{OH}$ corresponds to water oxygen pointing to the N-terminus. **e** Time-averaged water $\cos\theta_{OH}$ values. Water has larger anisotropy in the open$^{+4/+4}$ channel than in the closed$^{0/+1}$ channel. An orientation change is observed at residue A17 at low pH. **f** Time-averaged $\cos\theta_{OH}$ of channel water when the $+4/+4$ state is restrained to the high-pH solid-state NMR protein structure (PDB: 6PVR) and the $0/+1$ state is restrained to the low-pH solid-state NMR structure (PDB: 6PVT). Positive $<\cos\theta_{OH}>$ near W23 and negative $<\cos\theta_{OH}>$ near L8 are observed for the $+4/+4$ state despite the high-pH structure. Error bars for simulations are the standard error of the mean for four independent trajectories.

$^{13}$C-detected $^1$H spin-lock recoupling experiments conducted at lower temperatures produced identical recoupling curves (Supplementary Fig. 8b). Since chemical exchange and rotational motion are both slowed down at lower temperature, this data confirms that it is the charge state of histidine that controls water orientation anisotropy. Between the two histidines, H27 plays no role in controlling the water orientation, since no water anisotropy is seen near H27 in either $+1$ or $+4$ charge state (Fig. 5c–f and Supplementary Fig. 7). Thus, the orientational

order is induced by the spatially concentrated positive charge on the H19 tetrad.

**Water hydrogen bonds are less interrupted in the open channel than in the closed channel.** The presence of water orientational order within the open BM2 channel suggests that the water–water hydrogen-bonding network may be directional. To assess the possible influence of this directionality on proton transport, we counted the number of water–water hydrogen bonds and

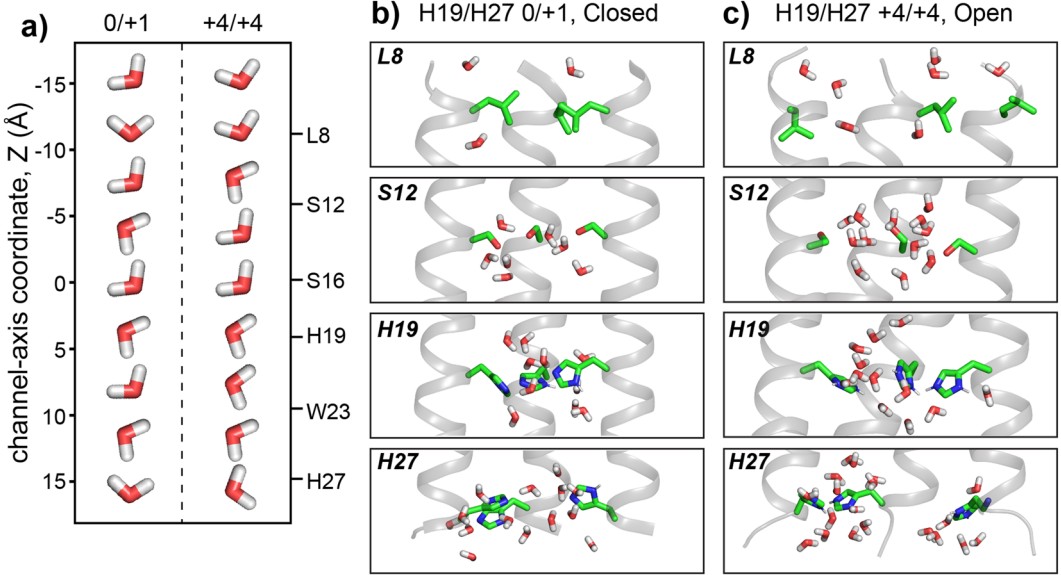

**Fig. 6 Water orientations in the closed$^{0/+1}$ channel and the open$^{+4/+4}$ channel. a** Most probable water orientations along the channel axis, plotted for 4 Å bins. **b**, **c** Representative snapshots of water in the **b** closed$^{0/+1}$ channel and the **c** open$^{+4/+4}$ channel. Only water within 3 Å of pore-facing residues are shown.

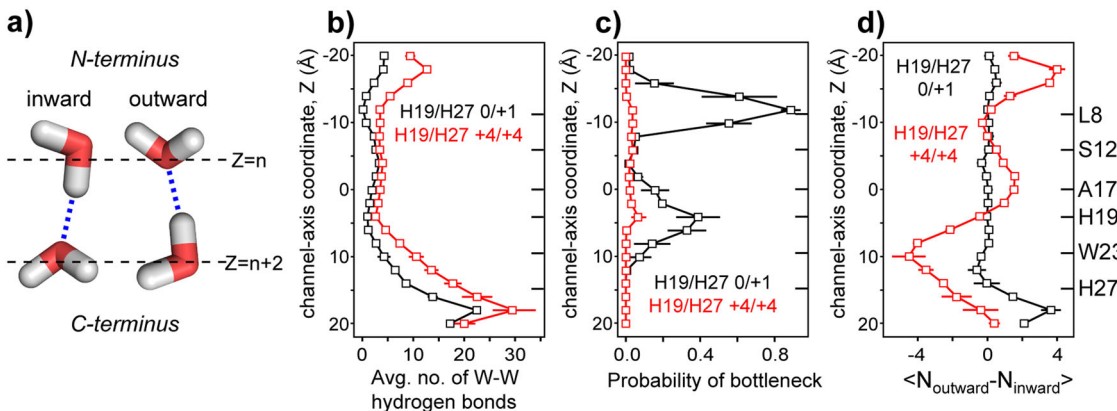

**Fig. 7 Water–water hydrogen bonding in the closed and open BM2 channels. a** Definition of the inward and outward water hydrogen bonds. **b** Average number of water–water hydrogen bonds along the channel axis. The open channel has more hydrogen bonds than the closed channel. **c** Probability of hydrogen-bond bottlenecks along the channel axis. The closed BM2 channel has high probabilities of bottlenecks at L8 and H19, while the open channel has low bottleneck probabilities throughout the channel. **d** Direction of water–water hydrogen bonds. The closed channel shows little preference for the hydrogen-bond polarity, while the open channel exhibits a bipolar hydrogen-bond orientation, reversed at H19. Error bars for simulations are the standard error of the mean for four independent trajectories.

evaluated their directions relative to the channel axis in the MD simulations (Fig. 7a, Methods section). We find that the open$^{+4/+4}$ state permits more water–water hydrogen bonds throughout the channel than the closed$^{0/+1}$ state (Fig. 7b). We classified hydrogen-bond bottlenecks as sites where connectivity along the channel was broken in an individual frame of the MD trajectory. Based on the statistics across 100 ns MD trajectories (Fig. 7c), we found two prominent bottlenecks, at L8 and H19, in the closed$^{0/+1}$ state, which are barriers to Grotthus-type proton transport. Bottlenecks are significantly more prevalent in the closed$^{0/+1}$ state, with at least one bottleneck in 97% of the frames, compared to only 24% of the frames in the open$^{+4/+4}$ state. On average, there are 3.7 bottlenecks in the closed$^{0/+1}$ state and 0.3 bottlenecks in the open$^{+4/+4}$ state. In addition, the water–water hydrogen bonds switch the directions in the open$^{+4/+4}$ state (Fig. 7d): the N-terminal half of the channel contains more

outward hydrogen bonds, while the C-terminal half contains more inward hydrogen bonds.

## Discussion

These solid-state NMR data and MD simulations show that the water dynamics, orientation, and hydrogen-bonding in the BM2 channel differs significantly between low- and high-pH states. Low-pH channel activation gives rise to (1) a larger pore size with two-fold more water, (2) faster water reorientation and chemical exchange on the ns–μs timescales, and (3) higher water orientational order and more directional hydrogen bonds. Using $^{13}$C detection of water $^1$H polarization, we selectively detected the small population of water inside the BM2 channel. The fact that the low-pH channel contains more water is not surprising, since the four-helix bundle is more loosely packed at low pH due to

electrostatic repulsion between the four positively charged H19 residues[27,28]. The excellent agreement between the measured and simulated number ratio of water molecules in the closed and open channels validates both the NMR and simulation methods. Further, the larger amount of water is not an artefact of the low temperature (277 K) at which the experiments were conducted, as shown by control simulations at 297 K (Supplementary Fig. 10).

Interestingly, the number of water molecules in AM2 channels determined by X-ray crystallography was found to be sensitive to the experimental conditions. Initial cryogenic and room-temperature synchrotron crystal structures of AM2 found comparable or more water in the high-pH channel than in the low-pH channel[11]. The number of waters was much higher in the cryogenic structures than the room-temperature structures. These results were subsequently attributed to water ordering at low temperature and radiation damage at room temperature. When crystal structures were obtained at room temperature using an X-ray free-electron laser (XFEL), more water was found in the low-pH AM2 channel than the high-pH channel[10]. Solid-state NMR measurements at moderate temperatures represent a non-perturbing approach for quantifying the amount of water in membrane-bound channels. Interestingly, the XFEL data of AM2 also indicate a doubling of water amount at low pH compared high pH, in good agreement with the current BM2 results. This similarity of acid-activated water increase of AM2 and BM2 channels implies that the conserved histidine dictates the water amount in the pore (vide infra).

At high pH, simulated water densities indicate low water occupancy at L8 and H19 (Fig. 2d) and frequent bottlenecks in the water hydrogen-bonding network (Fig. 7c). This hydrogen-bond disruption is not observed in AM2, which contains unbroken chains of water–water hydrogen bonds from the N-terminus to the histidine tetrad at both high and low pH[11]. We attribute this difference to the different oligomeric structures of AM2 and BM2: AM2 adopts a more open N-terminal vestibule at high pH that is closed after acid activation[24,57]. The interconversion between the two conformations underlies the functional asymmetry of the AM2 transporter, where protons are exclusively conducted inward. In comparison, the BM2 tetramer structure undergoes a symmetric scissor-like motion with respect to the center of the TM domain[27]. This conformational symmetry has been proposed to explain the ability of BM2 to conduct weak outward proton current in addition to strong inward current. The lower water amount at L8 in the closed$^{0/+1}$ BM2 channel is thus correlated with a more compact N-terminal pore. The minimum diagonal heavy-atom distance at L8 is 11 Å for the open$^{+4/+4}$ channel and 5.5 Å for the closed$^{0/+1}$ channel (Supplementary Fig. 1b). BM2's H19 p$K_a$'s have been measured and found to be significantly lower than the H37 p$K_a$ of AM2[29]: the average p$K_a$ of H19 is 5.1, while the average p$K_a$ of H37 in AM2 is 5.9. Although the peripheral histidine, H27, speeds up proton release from H19, even an H27A mutant of BM2 still manifests a low average H19 p$K_a$ of 5.6[28]. The current finding of a hydrogen-bond bottleneck at L8 provides a second mechanism of this depressed H19 p$K_a$'s, as the interruption of the water wire at L8 will slowdown proton relay from the N-terminus to H19.

Both NMR relaxation and MD simulations indicate that water in the low-pH channel reorients more rapidly than water in the high-pH channel (Fig. 4, Supplementary Tables 1 and 2, and Supplementary Movie 1). Despite this faster dynamics, water in the low-pH channel has higher anisotropy than water in the high-pH channel (Fig. 5), as shown by larger motionally averaged $^1$H CSA. Order parameters computed from simulations are in very good agreeement with the experimental data (Fig. 5c). The unexpected finding of faster water dynamics and higher water anisotropy can be rationalized by the fact that water molecules

that reorient quickly break and form hydrogen bonds with adjacent water molecules to permit proton exchange and thus Grotthuss hopping[5]. The higher water anisotropy at low pH is also fully consistent with the increased number and directionality of water–water hydrogen bonds at low pH (Fig. 7). This hydrogen bonding extends in opposite directions on either side of H19, with the C-terminal water favoring the inward hydrogen bonds and N-terminal water favoring outward hydrogen bonds. We attribute this hydrogen-bond polarity switch to the concentrated positive charge on the H19 tetrad.

The pH-dependent dynamics and orientations of BM2 channel water are surprisingly similar to the water properties in AM2 channels, despite the low sequence homology and the distinct oligomeric structures of the two proteins. AM2 also contains more dynamic water at low pH than at high pH, as manifested by picosecond time-dependent lineshapes in 2D IR spectra[58], longer water $T_2'$ relaxation times[25], and a larger number of half-occupancy water molecules in crystal structures[10]. MD simulations of AM2 also indicated that the water hydrogen-bond direction switches above and below H37[11,13]. The fact that both channels exhibit faster water dynamics and a switch in hydrogen-bond direction at the proton-selective histidine at low pH suggests that the proton-selective histidine, without assistance from other pore-lining residues, controls the water amount, water anisotropy, and water wire directionality in both AM2 and BM2 channels. The ps–ns water reorientation and the µs histidine ring reorientation serve to complete the hydrogen-bonded chain to enable proton transfer.

The fact that the tetrameric conformation of M2 plays a lesser role in channel-water properties than the selectivity-filter residue is also supported by MD simulations that used a coiled-coil structure of BM2 solved in detergent micelles[59,60]. Despite the significant structural differences from the bilayer-bound structure[27], the simulations reached the same conclusion about increased pore size and pore hydration at low pH. The charge-flipped MD simulations shown here lend further support to the essential role of the H19 charge state for regulating the water-wire properties. When the H19 tetrad is neutral, even if the four-helix bundle is loosened, the number of water molecules and the water orientational order are low. Conversely, when the H19 tetrad is highly charged, even if the four-helix bundle is tightened, the water amount and orientational order increase significantly (Fig. 5f and Supplementary Fig. 9). Therefore, the polarity of the hydrogen-bond network is electrostatically controlled by the charge state of the histidine. For AM2, the protein conformation and dynamics play the roles of limiting the net proton flux and dictating the directionality of the proton flux[24,25]. Whether proton flux in BM2 is rate-limited by protein conformational dynamics has not been shown, but the similar rates of proton conduction and conserved HxxxW motif suggest that this may also be the case for BM2. The millisecond motion of the four-helix bundle accounts for the ~1000 s$^{-1}$ proton flux of the M2 channels, while the symmetry of the helical motion affects the extent of the reverse proton current.

The combined measurements of water amount by polarization transfer NMR, water motional rates by relaxation NMR, and water orientation by recoupling NMR, represent a powerful approach for elucidating water properties in membrane proteins. MD simulations complement these NMR experimental results by providing site-specific information about water dynamics and hydrogen bonding. Our results show that proton conduction in the open state of BM2 is correlated with fast water rotational and translational diffusion, which optimizes the orientation of a hydrogen-bonded water wire[61,62]. The selectivity-filter histidine electrostatically controls water orientation and dynamics, and switches the water hydrogen-bond directions above and below

itself. The coexistence of dynamic disorder and orientational order of water may be a general property of channels and pumps that contain more than a single file of water molecules.

## Methods

**BM2 protein purification and membrane sample preparation.** Uniformly $^{13}$C and $^{15}$N labeled BM2 (residues 1–51) was produced as described previously[27]. The protein sequence corresponds to the influenza B/Maryland/ 1/2001 strain (MFEPFQILSI CSFILSALHF MAWTIGHLNQ IKRGVNMKIR IKGPNKETINR). This construct contains the TM domain and a portion of the cytoplasmic domain of full-length BM2[63]. BM2 (1–51) was expressed as a SUMO-tagged fusion protein in *E. coli* BL21 (DE3) cells and purified using nickel affinity chromatography. The tag was then cleaved using SUMO protease, and the native BM2 was purified using preparative reverse phase HPLC.

Two BM2 samples bound to 1-palmitoyl-2-oleoyl-sn-glycero-3-phosphoethanolamine (POPE) membranes at pH 7.5 and pH 4.5 were used for study. Nearly identical $^{13}$C and $^{15}$N chemical shifts were observed for BM2 transmembrane residues in POPE, 1-palmitoyl-2-oleoyl-sn-glycero-3-phosphocholine: 1-palmitoyl-2-oleoyl-sn-glycero-3-phospho-(1′-rac-glycerol) (POPC:POPG) and 1,2-dilauroyl-sn-glycero-3-phosphoethanolamine (DLPE) membranes[27]. Since the bulk of the structure determination of BM2 in lipid bilayers was done with POPE lipid membranes, the two samples studied herein relied on this membrane. The protein:lipid molar ratio was 1:16, the protein mass was 8–9 mg for each sample, obtained from 3 L expression. The protein was reconstituted into the POPE membrane by organic solvent mixing as described before.[27] A Tris buffer (20 mM Tris-HCl, 2 mM ethylenediaminetetraacetic acid (EDTA), 0.2 mM NaN$_3$, and 2 mM TCEP) was used for the pH 7.5 sample while a citrate buffer (20 mM sodium citrate, 2 mM EDTA, 0.2 mM NaN$_3$, and 2 mM TCEP) was used for the pH 4.5 sample. The proteoliposomes were ultracentrifuged, dried to ~40 wt% water, then spun into 3.2 mm MAS rotors for solid-state NMR experiments.

**Solid-state NMR experiments.** Solid-state NMR experiments were conducted on a Bruker Avance II 800 MHz (18.8 T) spectrometer equipped with a triple-resonance $^1$H/$^{13}$C/$^{15}$N Efree 3.2 mm magic-angle-spinning (MAS) probe. All experiments were carried out under 14 kHz MAS at a thermocouple-reported temperature of 263 K. The sample temperature was moderately higher due to frictional heating, and was estimated to be 273 K based on the water $^1$H chemical shift[64]. Two-pulse phase modulation (TPPM) $^1$H decoupling[65] was applied at an rf field of 71 kHz during $^{13}$C acquisition. $^{13}$C chemical shifts were referenced externally to the adamantane CH$_2$ chemical shift at 38.48 ppm on the tetramethylsilane scale and $^1$H chemical shifts were referenced to the Hγ peak of 1-palmitoyl-2-oleoyl-sn-glycero-3-phosphocholine (POPC) at 3.26 ppm on the tetramethylsilane scale[66].

$^{13}$C-detected $^1$H T$_2'$ experiments (Supplementary Fig. 2a) started with the selection of water $^1$H magnetization using a 1.5-ms Gaussian 90° excitation pulse, followed a variable echo period $\tau_{echo}$ containing a 7-μs 180° pulse in the middle. This $\tau_{echo}$ period was synchronized with the rotor period ($\tau_r$) and ranged from 0.14 ms to 7.14 ms. For $^1$H-detected T$_2'$ measurements, $\tau_{echo}$ ranged from 0 to 714.3 ms. The water $^1$H magnetization was transferred to protein with a spin diffusion mixing time $\tau_{SD}$ of 4 ms, followed by a CP contact time of 500 μs for $^{13}$C detection. To ensure that all $^1$H polarization detected originates from water, control experiments were conducted with a 0.1 ms $\tau_{mix}$. No appreciable signal was observed with this mixing time (Supplementary Fig. 2a), indicating that all peptide signals, including the Hα signal that can overlap with water, were suppressed. $^1$H spin diffusion buildup experiments used the same pulse sequence as the $^{13}$C-detected $^1$H T$_2'$ experiments, with the echo delay fixed at 0.28 ms.

$^{13}$C-detected water $^1$H R$_{1\rho}$ experiments (Supplementary Fig. 2b) started with water magnetization selection using a 1.5-ms Gaussian 90° excitation pulse. This was followed by a $^1$H spin-lock pulse for a variable delay $\tau_{SL}$ with field strengths ($\omega_1/2\pi$) of 3 kHz to 71 kHz. A total of 16 rf field strengths were used, and $\tau_{SL}$ values ranged from 1 ms to 15 ms for each field strength. The water $^1$H magnetization was transferred to protein with a $\tau_{mix}$ of 4 ms, followed by a CP contact time of 500 μs. To ensure that all $^1$H polarization originates from water, control experiments were run with a 10-μs $\tau_{mix}$ and 1 ms spin-lock period. No appreciable signal was observed under these conditions, indicating that all peptide signals were suppressed and all $^1$H polarization originates from water.

For spin-lock recoupling experiments (Supplementary Fig. 2c)[51], the $\omega_1/2\pi$ values were 7 kHz, 14 kHz, and 28 kHz, corresponding to matching conditions of $\omega_1 = 0.5 \omega_r$, $\omega_r$, and $2\omega_r$, and the $\tau_{SL}$ values ranged from 0.05 ms to 10 ms. The spin-lock field strength was optimized to achieve the best dephasing (lowest intensity) at a mixing time of 1 ms, which corresponds to the observed minimum for the $\omega_r$ condition and the initial fast decay for the $2\omega_r$ condition (Supplementary Fig. 7). For the $^{13}$C-detected RECRR experiments (Supplementary Fig. 2d), the entire spin-lock period was separated into two phase-inverted portions, each containing a 180° pulse in the middle. The total $\tau_{SL} = 4n\tau_r$, where $n\tau_r$ is the duration of each sub-block. We used $\tau_{SL}$ values of 0.28–10.86 ms. The water $^1$H magnetization was transferred to protein with a mixing time $\tau_{mix}$ of 4 ms, followed by a CP contact time of 500 μs. These RECRR experiments were conducted only at

spin-lock field strengths of 7 kHz, 14 kHz, and 28 kHz, using the same optimized field strengths as for the spin-lock recoupling experiments.

**T$_2'$ and T$_{1\rho}$ relaxation analysis.** All spectra were processed in TopSpin (Bruker Biospin). NMR data analysis and fitting were conducted using MATLAB and OriginPro. $^1$H- and $^{13}$C-detected T$_2'$ relaxation curves were fit to a biexponential function of the form:

$$I(\tau) = p_1 e^{-\tau/T_{2,1}'} + p_2 e^{-\tau/T_{2,2}'} \tag{1}$$

The two time constants and associated populations are reported in Supplementary Table 1. All error bars reflect a 68% confidence interval.

The rotational correlation time $\tau_{rot}$ is obtained from the water $^1$H T$_2'$ values using the Bloembergen−Purcell−Pound (BPP) theory[38] as follows:

$$\frac{1}{T_2} = \frac{3\mu_0^2\gamma^4\hbar^2}{320\pi^2 r^6}\left(3\tau_c + \frac{5\tau_{rot}}{1+\omega_0^2\tau_{rot}^2} + \frac{2\tau_{rot}}{1+4\omega_0^2\tau_{rot}^2}\right) \tag{2}$$

Here $\mu_0$ is the magnetic permeability of free space, $\gamma$ is the $^1$H gyromagnetic ratio, $\hbar$ is the reduced Planck's constant, $r$ is the intramolecular $^1$H–$^1$H distance of water (1.58 Å), $\tau_{rot}$ is the rotational correlation time, and $\omega_0$ is the proton Larmor frequency which is $2\pi \times 800 \times 10^6$ rad/s in a magnetic field of 18.8 T.

$^{13}$C-detected water $^1$H R$_{1\rho}$ decays were fit to variable-amplitude single exponential functions. The R$_{1\rho}$ rates at $\omega_1$ values away from the three recoupling conditions ($\omega_1 = 0.5\omega_r$, $\omega_r$, and $2\omega_r$) were used for the relaxation dispersion analysis. These R$_{1\rho}$ values were fit to Eq. (3), which assumes a two-state exchange model[40,42]:

$$R_{1\rho} = R_{1\rho}^0 + \frac{\phi_{ex}(1/\tau_{ex})^2}{\omega_1^2 + (1/\tau_{ex})^2} \tag{3}$$

Here $R_{1\rho}^0$ is the rotating-frame spin-lattice relaxation rate in the limit of infinitely strong $\omega_1$. The exchange parameter $\phi_{ex} = p_1 p_2 \Delta\omega^2$ depends on the populations of the two states, $p_1$ and $p_2$, and on the isotropic chemical shift difference $\Delta\omega$ between the two states. $\tau_{ex}$ is the exchange time constant and $\omega_1$ is the rf spin-lock field strength. Fitting the R$_{1\rho}$ dispersion profile to Eq. (3) requires that the relaxation dispersion is solely due to fast exchange between two different isotropic chemical shifts.

**Numerical simulations of spin-lock recoupling to extract $^1$H chemical shift anisotropy.** The $^{13}$C-detected water $^1$H R$_{1\rho}$ rates at the three recoupling conditions cannot be fit to an exponential function because they exhibit an additional fast oscillating component due to coherent contributions. We simulated these curves using the SpinEvolution software[67]. The simulations utilized an eight-spin system consisting of the protons in four hydrogen-bonded water molecules (Supplementary Fig. 6a)[68]. Numerical simulations were run using the NMR experimental conditions, including a static magnetic field of 18.8T (800 MHz), an MAS frequency of 14 kHz, and a $^1$H carrier frequency set to be on-resonance with the water protons. These simulations began with $^1$H magnetization along the x-axis, and the amount of magnetization was monitored with increasing spin-lock times (Supplementary Fig. 6b). Motionally averaged dipolar order parameters $S_{HH}$ between the two water protons were varied by setting the dipolar couplings to be 5% of the rigid-limit coupling. Motionally averaged $^1$H CSA was varied and compared with the rigid-limit water $^1$H CSA of 28.5 ppm[53]. The results of the simulations were found to be insensitive to the $^1$H CSA tensor orientation, thus we set $\alpha = \beta = \gamma = 0°$. We note that when $^1$H CSA is included but $^1$H–$^1$H dipolar couplings are turned off, the eight-spin simulation reverts to a single-spin situation.

The water $^1$H spin-lock recoupling data exhibit oscillations that are significantly dampened compared to what is expected for a single CSA magnitude subjected to homogeneous B$_1$ irradiation (Supplementary Fig. 6c). This dampening is likely due to a combination of incoherent R$_{1\rho}$ relaxation, B$_1$ field inhomogeneity, and a distribution of $^1$H CSAs. We are unable to differentiate the effects of B$_1$ field inhomogeneities from the effects of CSA distribution. Thus, we assume that there is a single dominant motionally averaged CSA. To fit the measured spin-lock recoupling profile, the B$_1$ inhomogeneity for a $^1$H loop-gap resonator[69] was modeled as such: CSA experiments were simulated for rf spin-lock field strengths corresponding to 90–100% in 0.2% increments of the recoupling conditions $0.5\omega_r$, $\omega_r$, and $2\omega_r$. We note that an rf field inhomogeneity of 110-100% results in approximately the same dephasing curve as an rf inhomogeneity of 90–100%, e.g., dephasing is roughly symmetric about the matching condition. These were then multiplied by a polynomial weighting function, $(\Delta\omega_1/\Delta\omega_{1,max})^4$ to account for the fact that the majority of the sample experiences the desired spin-lock field strength, while the contribution from spin-lock fields that are 10% different from the desired value is relatively small (Supplementary Fig. 6d). The weighted coherent decay curves were summed and normalized to one, before being multiplied with an exponential decay function with a time constant of 5 ms to account for the incoherent R$_{1\rho}$ relaxation components. The RMSD between the measured and simulated spin-lock recoupling curves was calculated to find the best-fit motionally averaged CSA for each recoupling curve (Supplementary Fig. 6e).

**System setup and molecular dynamics simulation protocols**. Initial systems for the MD simulations were set up using CHARMM-GUI[70] using the following steps. First, pore waters were added to the lowest-energy NMR structures (residues 1-33) of the closed (6PVR) and open (6PVT) BM2 channels[27], using the DOWSER package[71], which omitted a drain step to prevent the dehydration of the cavity. The C-terminal residue R33 was capped with a methyl group and hydrogens were added using CHARMM-GUI[70]. The charge states of H19/H27 were assigned as 0/+1 for the high-pH helix bundle and +4/+4 for the low-pH helix bundle. Next, the prepared BM2 structures, including the pore water molecules, were inserted into an 80 Å × 80 Å POPE bilayer containing 190 lipids, orienting the channel-axis perpendicular to the membrane plane. The bilayer was solvated with a 30-Å-thick water layer on each side of the bilayer with 150 mM NaCl. The boundaries of the simulation systems were handled using periodic-boundary conditions. The system was energy-minimized using 5000 steps of steepest descent and equilibrated at 297 K and 1 atm. The equilibration protocol consisted of a series of short simulations (375 ps in total) with harmonic position restraints on the protein backbone (BB), protein sidechains (SC), lipid headgroups (LH), and lipid tails (LT; harmonic restraints as specified in Supplementary Table 3). The equilibrated structures were next subjected to four independent production runs of 130 ns in an NPT ensemble, with semi-isotropic pressure coupling at 1 atm, using a Parrinello-Rahman barostat[72,73] and 277 K, using the V-rescale thermostat[74]. This temperature was chosen to mimic the experimental temperature of ~273 K. All water analysis was performed on the last 100 ns of MD trajectories, leaving out the first 30 ns of unconstrained NPT simulation as equilibration. All simulations were carried out using GROMACS 2018.3[75] and the CHARMM36 forcefield, using the TIP3P water model. Short ranged non-bonded interactions employ a 1.2-nm cut-off, long-range electrostatics were calculated using the smooth particle mesh Ewald (PME) method. The LINCS algorithm[76] was used to constrain h-bonds and to enable stable integration using a 2 fs timestep, simulation snapshots were saved every 10 ps for analysis.

Although the TIP3P water model has been used for peptide and small protein simulations below room temperature[77,78], we note that there are known limitations on the accuracy of TIP3P at 277 K[79]. Therefore, we performed control simulations at 297 K to reproduce the analysis (Supplementary Fig. 10). The simulations at 297 K yield qualitatively comparable results to those at 277 K. Importantly, we still find increased water density (Supplementary Fig. 10a, b), water dynamics (Supplementary Fig. 10c, d), water order (Supplementary Fig. 10e–g), and hydrogen bonding (Supplementary Fig. 10h–j) in BM2 at pH 4.5 compared to BM2 at pH 7.5.

While most water analysis was conducted on unrestrained simulations, additional simulations were carried out where the protein structure was restrained to the solid-state NMR backbone structure using harmonic restraints (50 kJ mol$^{-1}$ nm$^{-2}$ force constant). Specifically, control simulations were performed in which the open BM2 (+4/+4 H19/H27 charge state) was kept near its NMR structure (PDB: 6PVT) and the closed BM2 (0/+1 H19/H27 charge state) was kept near its NMR structure (PDB: 6PVR). These restrained simulations had dilated channels compared to the unrestrained simulations (Supplementary Fig. 1b), thus giving a larger number of channel-water molecules and slightly increased translational and rotational dynamics. However, the ratio of water molecules between the +4/+4 and +0/+1 H19/H27 charge states, the increased water mobility for the +4/+4 charge state, and the orientation anisotropy for the +4/+4 charge state, were preserved (Supplementary Fig. 4a). Additional restrained simulations that investigate the effect of the H19/H27 charge state on water-orientation, in the absence of large-scale conformational changes in BM2, (Fig. 5f and Supplementary Fig. 9) restrain the open BM2 to its NMR structure (PDB: 6PVT) while using the closed BM2 charge state (0/+1 H19/H27), and restrain the closed BM2 to its NMR structure (PDB: 6PVR) while using the open BM2 charge state (+4/+4 H19/H27).

**Analysis of the MD channel-water trajectories**. The water correlation times, orientation, and hydrogen bonding in the last 100 ns of each simulation were analyzed using the MDAnalysis package[80]. The scripts for calculating these quantities from the MD trajectories are available via github (https://github.com/mjmn/BM2-MD). Reported values and error bars represent the mean and standard error of the mean, over the independent trajectories included in calculating the reported quantity.

**Channel-axis coordinate, Z**. The channel-axis coordinate, $Z$, is defined as the projection onto a vector along the principal axis of the Cα atoms in the BM2 channel, with the vector origin set to the center of the four A17 Cα atoms, and the positive direction oriented towards the C-terminus of BM2 (channel-axis vector). To assign $Z$ values to residues, we use the average $Z$ value of all heavy atoms in the corresponding residue over the combined trajectories (Supplementary Table 4).

**Water density**. Water density is reported as the average number of water molecules within a 4-Å bin along the Z axis. Water molecules are assigned based on the position of their oxygen atom. Only water molecules that are within 15 Å of the channel-axis vector are included. Restraining the protein backbone to the solid-state NMR structure resulted in slightly more water within both the open and closed channels, but the ratio between the amount of water in each channel

remained the same (Supplementary Fig. 4a). This is due to the fact that the equilibrated structures are somewhat constricted compared to the solid-state NMR structures (Supplementary Fig. 1b).

**Water rotational dynamics**. Rotational auto-correlation functions are calculated as

$$C_\mu(\tau) = \frac{1}{2}\langle 3\left[\boldsymbol{\mu}(t)\cdot\boldsymbol{\mu}(t+\tau)\right]^2 - 1\rangle_t \tag{4}$$

where $\boldsymbol{\mu}(t)$ is the orientation of the water-molecule dipole vector at time $t$. Fitting the long-time decay (>10 ps) of the auto-correlation function to an exponential function gives the rotational correlation time $\tau_{rot}$:

$$C_\mu(\tau) \propto e^{-\tau/\tau_{rot}} \tag{5}$$

The $\tau_{rot}$ value was determined as a function of the displacement along the channel axis by assigning each water molecule to the channel-axis coordinate bin (4 Å width) at which its oxygen atom resides at $\tau = 0$.

**Water translational dynamics**. The translational dynamics of water is quantified by fitting the mean-square displacement, MSD,

$$\text{MSD}(\tau) = \langle\left|\mathbf{r}(t+\tau) - \mathbf{r}(t)\right|^2\rangle_t \tag{6}$$

to the following function:

$$\text{MSD}(t) = 6Dt^\alpha \tag{7}$$

where $\boldsymbol{r}(t)$ is the position of a water oxygen at time $t$, and $\alpha$, and $D$ are parameters that define the translational motion of water. In the BM2 channel, the water dynamics were found to be sub-diffusive (i.e., $\alpha < 1$). This quantity was determined as a function of the displacement along the channel axis.

The flux of water molecules across the channel was calculated from both the N-terminus and the C-terminus by counting the number of water molecules that enter the channel on one side and exit on the other side. A water molecule contributes to the flux at the time it exits the channel. For a given water molecule, it is labeled as: entering the channel on the N-terminal side when it crosses the two-dimensional plane defined by $Z = -20$ Å in the positive $Z$ direction, exiting the channel on the N-terminal side when it crosses the $Z = -20$ plane in the negative $Z$ direction, entering the channel on the C-terminal side when it crosses the two-dimensional plane defined by $Z = 20$ Å in the negative $Z$ direction, and exiting the channel on the C-terminal side when it crosses the $Z = 20$ Å plane in the positive $Z$ direction. No significant difference in $D(Z)$ was found between the closed and open channel. The slowdown in translational dynamics in the closed channel, summarized here using only the coefficient, is also apparent from MSD versus time plots (Supplementary Fig. 4b) and in the total number of transported water molecules through the channel: this number is 2.73 ± 0.41 molecules/ns in the open channel and 0.08 ± 0.05 molecules/ns in the closed channel (Fig. 3g).

**Water orientation**. The water orientational order in the channel was determined by analyzing the probability distribution of the dot product between the water OH-bond vector and the channel-axis vector, $Z$. The full probability distributions, $P(\cos(\theta_{OH}))$, are shown in Supplementary Fig. 7. Several quantities derived from these probability distributions are presented in the main text and are described here.

We used the average value $\langle\cos(\theta_{OH})\rangle$ to assess the orientational preference of water along the channel axis (Fig. 5e). This quantity is defined as:

$$\langle\cos(\theta_{OH})\rangle(Z) = \int_{-1}^{1}\cos(\theta_{OH})P(\cos(\theta_{OH})|Z)d\cos(\theta_{OH}) \tag{8}$$

To compare with the measured order parameters more directly, we also calculated the OH-bond vector order parameter according to:

$$S_{OH}(Z) = \frac{1}{2}\int_{-1}^{1}\left[3\cos^2(\theta_{OH}) - 1\right]P(\cos(\theta_{OH})|Z)d\cos(\theta_{OH}) \tag{9}$$

For Fig. 5e, the probability distribution of the $\cos(\theta_{OH})$, $P(\cos(\theta_{OH})|Z)$, was determined for water molecules within a 4-Å bin along the channel axis. To directly compare with the measured NMR orientations, which do not have spatial resolution along the channel axis, we also determined the probability distribution for all water molecules from $-12$ Å to $+12$ Å along the channel coordinate.

We also analyzed non-uniformity in the $P(\cos(\theta_{OH}))$ distribution (Supplementary Fig. 4d) using an entropy parameter ($\Delta\Gamma$) defined as

$$\Delta\Gamma = \sum_{i=0}^{N_{bin}} P(\cos\theta_{OH})\ln[P(\cos\theta_{OH})] + \ln[N_{bin}] \tag{10}$$

where $N_{bin}$ is the number of bins used to discretize the $P(\cos(\theta_{OH}))$ distribution, and the $\ln[N_{bin}]$ offset is applied such that $\Delta\Gamma = 0$ for a uniform probability distribution. Unlike $\cos(\theta_{OH})$ and $S_{OH}$, $\Delta\Gamma$ does not depend on the direction of the reference vector for calculating $\cos(\theta_{OH})$. Higher $\Delta\Gamma$ values correspond to higher orientational preference. Upon restraining the protein backbone to the solid-state NMR structure, nearly identical anisotropy parameters were observed as in the unrestrained simulations (Supplementary Fig. 4a).

**Water–water hydrogen bonding**. Water hydrogen bonds were analyzed with the MDAnalysis package[80]. Hydrogen bonds were defined based on default geometric criteria in the package (https://www.mdanalysis.org/docs/documentation_pages/analysis/hbond_analysis.html). Specifically, a water–water hydrogen bond involved a separation of 3.0 Å or less between one water molecule's hydrogen atom and a second water molecule's oxygen atom, and an angle of 120° or more between the oxygen bound to the hydrogen-bonded hydrogen atom, the hydrogen-bonded hydrogen atom, and the acceptor oxygen atom.

A frame stride of 10 was used. The network analysis is carried out for each frame by assigning relevant water molecules in that frame to channel axis slices and then identifying hydrogen bonds across channel-axis slices in that frame. For each frame, each water molecule's channel-axis coordinate for that frame is determined by projecting the water molecule's oxygen atom onto the channel axis. If the water molecule is within the channel-axis coordinate limits of $-20$–$22$ Å, and is within 15 Å of the channel-axis vector, it is included in the analysis for that frame. Water molecules are binned in 2-Å-thick bins along the channel axis.

We counted the water–water hydrogen bonds in the simulated trajectories between water molecules assigned to bins separated by 2 Å, in which one water molecule is one slice ($Z = n$) and the second water molecule is in the adjacent slice closer to the C-terminus ($Z = n + 2$; Fig. 7a). For each bin, all hydrogen bonds associated with that bin are those between water molecules in that channel-axis coordinate bin ($Z = n$) and those in the bin 2 Å closer to the C-terminus ($Z = n + 2$). A water–water hydrogen bond is defined as inward, or a donor, when an N-terminal water molecule points its hydrogen atom towards a C-terminal water oxygen atom. A water–water hydrogen bond is defined as outward, or an acceptor, when an N-terminal water oxygen atom receives a hydrogen bond from a C-terminal water hydrogen atom (a C-terminal water molecule points its hydrogen atom towards an N-terminal water oxygen atom, as shown in Fig. 7a). For instance, the hydrogen-bond count associated with the channel-axis coordinate 4.0 Å bin represents the count of hydrogen bonds between the water molecules in the 4.0-Å bin and those in the 6.0-Å bin. The number of outward (acceptor) hydrogen bonds associated with the 4.0-Å coordinate represents the count of hydrogen bonds between water molecules in the 4.0-Å bin and those in the 6.0-Å bin in which the water molecule in the 4.0-Å bin is the acceptor and the water molecule in the 6.0-Å bin is the donor. An inter-slice outward-inward (or acceptor-donor) difference is computed as the difference for each frame for each slice between the inter-slice hydrogen-bond count in which the water molecules in the slice act as acceptors (in water–water hydrogen bonds with the adjacent slice 2 Å closer to the C-terminus) and the inter-slice hydrogen-bond count in which the water molecules in the slice act as donors (in water–water hydrogen bonds with the adjacent slice 2 Å closer to the C-terminus). Bottlenecks are considered to be present in a frame for a slice when there are zero hydrogen bonds between the water molecules in that slice and water molecules in the slice 2.0 Å further toward the C-terminus. For each frame, the total count of inter-slice bottlenecks along the entire channel axis is recorded. Uncertainty is quantified as the standard error of the mean using the averages for the four replicates (as described above) using scipy.stats.sem[81].

**Statistics and reproducibility**. Sufficient number of scans were used for all NMR experiments to obtain good signal-to-noise. All error bars for NMR analyzed parameters reflect a 68% confidence interval. For MD simulations, the reported values and error bars represent the mean and standard error of the mean, over the four independent trajectories included in calculating the reported quantity.

**Reporting summary**. Further information on research design is available in the Nature Research Reporting Summary linked to this article.

## Data availability

The data that support the findings of this study are available from the corresponding author upon reasonable request. Source data underlying plots shown in figures are provided in Supplementary Data 1.

## Code availability

The code and scripts used for the MD portion of this study are available at https://github.com/mjmn/BM2-MD.

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

## Acknowledgements

This work is supported by National Institutes of Health grant GM088204 to M.H. and National Science Foundation grant CHE-1654415 to A.P.W.; D.A.S. is grateful to the Fannie and John Hertz Foundation and the National Science Foundation Graduate Research Fellowship program (1122374) for fellowships. M.D.G. is supported by an NIH Ruth L. Kirschstein Individual National Research Service Award (1F31 AI133989).

## Author contributions

M.H. designed the project. M.D.G., V.S.M., and A.J.D conducted the solid-state NMR experiments, analyses, and NMR numerical simulations. M.J.M.N., D.A.S. and A.P. W. performed the MD simulations and analyses. All authors discussed and interpreted the data and wrote the manuscript.

## Competing interests

The authors declare no competing interests.
