## [Peer Review File · Communications Biology]

Reviewers' comments:

Reviewer #1 (Remarks to the Author):

A. Summary of the key results

The authors made use of solid-state NMR spectroscopy and molecular dynamics (MD) simulations to characterize the behavior of water in the influenza B M2 protein channel in the low-pH open channel and in the high-pH closed channel.

[1] The number of water molecules contained in the low-pH open channel is approximately the double of that contained in the high-pH closed channel.

[2] Water molecule exhibits rotational motion, which is faster in the low-pH open channel, and subdiffusive dynamics, which is accentuated in the high-pH closed channel. The water flux through the pore is different between the structures, being greater in the low-pH open channel.

[3] H19 and H27 histidine residues cause water to exhibit preferred orientation, with H19 playing a major role in control the orientational order.

B. Originality and significance: if not novel, please include reference.

The study is novel in the sense that the authors report extensive experimental and computational work on influenza B M2 pore channel embedded in a POPE membrane. However, two previous computational papers cited by the authors focus on the role of specific residues as histidines and serines for the channel functionality (see papers by Q.-C Zheng and Q. Zheng cited by the authors, Ref 47 and 48).

The experimental work is remarkable; the computational work has serious shortcomings, and I would not include it -see all my reservations in what follows.

C. Data & methodology: validity of approach, quality of data, quality of presentation

The authors do not justify the choice of using 1-palmitoyl-2-oleoyl-sn-glycero-3-phosphoethanolamine (POPE) lipid molecules for building the lipid bilayer. This type of lipid molecule is also used for their experimental work but an explanation for this choice is missing. Please, specify the software used to build the lipid bilayer (maybe CHARMM-GUI?).

The statement "The bilayer was solvated by a 30 Å thick water layer on each side of the bilayer with 150 mM NaCl." is repeated twice (line 3 and 6 of System setup and molecular dynamics simulation protocol paragraph).

Please specify the force field used for water – being the study focused on water, it is one of the most important pieces of information, if the calculations need to be reproduced. Also, in this work the choice of using an appropriate water model is crucial because the simulations are carried out at the maximum density temperature (TMD), below room temperature (see C. Vega and J. L. F. Abascal, Phys. Chem. Chem. Phys., 2011, 13, 19663–19688).

Concerning the same topic, it would be important to report a reference of protein simulations carried out with the same force field at low temperature.

According to the protein data bank website, the (6PVR) closed and (6PVT) open BM2 structures were

determined experimentally at 285 K. Please, justify the choice of carrying out a very short equilibration (375 ps in total) at 297 K without performing a heating procedure and the choice of carrying out simulations at 277 K without performing a descendent temperature ramp procedure. Being the simulations carried out at low temperature, the system would need a long equilibration procedure.

The number of steps used for the steepest descent energy-minimization protocol should be written in the main text or there should be a reference for the Supplementary Information. These details of the simulation setup procedure are missing in the text and are important in case the calculations need to be reproduced: the simulation time step, the saving trajectory time step, the values of the harmonic position restrains applied to the protein backbone, protein sidechains, lipid headgroups, and lipid tails. Also, are periodic boundary conditions applied? Is a switching function with a cut off used for the short-range interactions? Which is the time step used for integrating the equations of motion? The simulation time step is important also in terms of the simulation time resolution for computing water autocorrelation functions.

Please, specify how bonds involving hydrogen atoms are treated during the simulation, i.e. if the SHAKE or any other algorithm is used, and how electrostatic interactions are calculated (supposedly using smooth particle-mesh Ewald summation method). Given the importance of the electrostatics and the hydrogens in proton transport these details of the simulation setup are relevant.

The authors state in the System setup and molecular dynamics simulation protocol paragraph that 'While most water analysis was conducted on unrestrained simulations, additional simulations were carried out where the protein structure was restrained to the solid-state NMR backbone structure using harmonic restrains' without describing which structure (open or closed) is being restrained and the values of the harmonic restrains. Some information is written in the legend of Fig S1d, but all information should be reported in the Methods clearly.

The authors provide a movie without the reference to it in the main text.

The authors state: 'The equilibrated structures were next subjected to four independent production runs of 130 ns in an NPT ensemble, with semi-isotropic pressure coupling at 1 atm and 277 K.' Please, specify the algorithms used to maintain the pressure and temperature constant and the reference to the corresponding paper if any.

Please specify whether the average reported in table S4 for assigning the values of the channel-axis coordinates Z is calculated from the initial structure atom coordinate or from the equilibrated structure.

Data reported in Fig S4a for the unrestrained simulations should be reported in scattered format not with a dashed line and with their error bars for allowing proper comparison with the restrained simulation data.

The authors define accurately and correctly how they quantify the water translational dynamics and the water flux through the channels in the Methods. However, they are inaccurate when they describe their results in the main text (p.7 'We also examined water translational diffusion in the simulations, by tracking the number of water molecules that passed through the channel in either direction' and 'We next computed the mean-squared displacement (MSD) of water molecules in the channel (Fig. S4b),...'). Authors should maintain the definition of flux in the main text or consider using a collective diffusion model (see for instance F. Zhu, E. Tajkhorshid, and K. Schulten, PRL 93, 224501, 2004) if they discuss the translational diffusion through pores in terms of water permeation through channels.

The authors state: 'The number of water molecules increases the most near the hydrophobic valve formed by L8,...', which seems odd given the hydrophobic character of leucine residues. This could be due to the dimension/radius of the pore or to the pore electrostatics. It would be interesting to investigate further.

In Figure 2d the data at the channel-axis coordinate Z values of approximately -20, 15, and 20 are missing, whereas the relative ratio between those values is reported in Fig 2e. The plot in Fig 2d should contain all data or be displayed with all data in the Supplementary Information or the author should justify their choice of not showing such data.

In Fig 5e,f the authors report the results of the analysis of the water orientational order performed on the unrestrained and restrained simulations. The fact that by inverting the histidine charges between the closed and open structures, the original orientation of the dipoles is not recover completely (closed0/+1 differs from open0/+1 structure) indicates that there are more factors than the histidine charges influencing the dipole direction. These factors could be the electrostatic potential of the whole channel or system, or the geometry of the channel. Without exploring these factors, it is hard to demonstrate that the dipole alignment depends only on those charges.

D. Appropriate use of statistics and treatment of uncertainties Appropriate use.

E. Conclusions: robustness, validity, reliability

The authors state: ' We attribute the fast decay in ~ 10 ps to thermal motion of bulk-like water, and the slow decay to collective dynamics of the channel-water hydrogen-bonding network in the confined pore and interactions of water with protein residues'. The reference to "collective dynamics" is inaccurate and needs a better explanation. The rotational auto-correlation function as defined in the Methods provides information on the rotational motion of the dipole vector of a water molecule. Then, the function is calculated over all molecules and averaged, but it does not report on collective dynamics, which can be computed differently by means of MD simulations (see for instance S. Capponi, S. H. White, D. J. Tobias, M. Heyden, *J. Phys. Chem. B* 123 (2), 480-486, 2018; V. C. Nibali, G. D'Angelo, A. Paciaroni, D. J. Tobias, M. Tarek. *J. Phys. Chem. Lett.*, 5, 1181–1186, 2014; M. Tarek and D. J. Tobias, *PRL*, 89, 275501, 2002).

In the same statement about rotational dynamics, the authors wrote: 'We attribute the fast decay in ~ 10 ps to thermal motion of bulk-like water'. The characteristic rotational relaxation time of bulk-like water at room temperature is ~ 1 ps. With decreasing temperature, this characteristic time becomes slower, but without a proper discussion about how temperature affects rotational dynamics (simulations are carried out at 277 K) and about the water force field it is difficult to assign the fast decay of ~ 10 ps to thermal motion of bulk-like water. Also, at a given temperature water in confined geometries shows slower dynamics compared to bulk- like water (Teixeira, J., Zanotti, J.-M., Bellissent-Funel, M.-C., Chen, S.-H. *Phys. B* 1997, 234–236, 370; Bellissent-Funel, M.-C. *J. Phys.: Condens. Matter* 2001, 13, 9165; Starr, F. W., Nielsen, J. K., Stanley, H. E. *Phys. Rev. Lett.* 1999, 82, 2294.). Authors should comment on that because water in BM2 is confined.

Please provide a reference or an explanation about the electrostatic repulsion on this statement: '... the four-helix bundle is more loosely packed at low pH due to electrostatic repulsion of the +4 H19 tetrad.'

This statement is unclear and it should be discussed appropriately given the temperature at which the simulations are carried out: ' Further, this agreement suggests that the difference in water occupancy in the channel is insensitive to the effects of thermal fluctuations'

The authors state: 'The lower water amount at L8 in the closed^{0/+1} BM2 channel is therefore directly correlated with a compact N-terminal pore.'. It would be useful to have quantitative data or a reference to describe what "compact" means, as for instance the dimension of the radius of the pore at L8 site in the closed and open structure.

The authors state: 'Therefore, the polarity of the hydrogen-bond network is electrostatically controlled by the charge state of the histidine. In comparison, the protein conformation and dynamics play the roles of limiting the net proton flux and dictating the directionality of the proton flux.' Further analysis on the conformation and shape of the pore and on the dynamics of the protein, i. e. of the residues and side chains lining the pore, should be carried out to support this statement.

The author state: 'Our results show that proton transport by BM2 requires fast water rotational and translational diffusion that optimizes the orientation of a hydrogen-bonded water wire.' In order to make this conclusion plain and comprehensible, the authors should provide details or references to explain how the rotational and translational diffusion of water are related to the proton transport and how the results presented in this work fit into the big picture.

F. Suggested improvements: experiments, data for possible revision

Would it be possible to provide a figure representing the low-pH and high-pH BM2 structures overlapping? It would help the reader to understand better the differences between the open and closed structure at the beginning and at the end of the simulation and the size and geometry of the cavity without looking at the structure paper (Mandala, V.S., Loftis, A.R., Shcherbakov, A.A., Pentelute, B.L. & Hong, M. *Nat. Struct. Mol. Biol.* 27, 160-167 (2020)).

The authors discuss extensively how temperature affects the experiments carried out on the AM2 channels. A discussion on how temperature might affect the experiments on the BM2 channels should be provided as well. Given the variability of the performance of water model and the computational complexity of describing the interactions between water and protein at low temperature with such models, it would be useful to perform simulations at room temperature and discuss the results.

G. References: appropriate credit to previous work?

References to computational work related to the setup of the simulations should be included along with adding the missing details which I underlined in the previous comments (part A). The rest of the work gives appropriate credit to previous study.

H. Clarity and context: lucidity of abstract/summary, appropriateness of abstract, introduction and conclusions

The abstract and the introduction are clearly presented. The authors should spend few more words in the introduction on the difference between influenza A M2 protein and influenza B M2 protein to guide properly the reader unfamiliar with these systems.

Reviewer #2 (Remarks to the Author):

This is a highly interesting manuscript. It describes the properties of water in an ion channel (M2) in two different states, an open state obtained at low pH, and a closed state obtained at high (neutral)

pH. This work build upon beautiful work by the Hong group on M2, including high-resolution structures of these two states (NSMB, Mandala et al 2020). It combines solid- state NMR with MD simulations and shows, for example, that

- the open state contains about 2-fold more water than the closed one
- water has different dynamics in the two states, leading to different (>30-fold) water transport
- there are “bottleneck” positions along the channel for water transport

The study used a number of ssNMR techniques to decipher the properties of water. In many cases, the story appears coherent, and there are quantitative matches between properties obtained by NMR and those obtained by MD simulations.

I think that this story will interest many researchers in different fields (channels; NMR; hydration dynamics), and thus I recommend that at some point it gets published. Among the elegant points of the paper (also worth being in the public domain) is the derivation of spin diffusion rates in the SI text.

This being said, from an NMR point of view it contains so many open points, where I think that the authors have taken shortcuts that must be investigated again. Briefly, I think that for almost every NMR observable there are either severe experimental pitfalls or even theoretical issues, that I am not sure that the conclusions really derive from the data in an unambiguous manner.

While I don't think that this is the case – I know Mei Hong's rigorous work for long time – it might even be that many of the matches are fortuitous. In any case, the interpretation of a number of parameters really needs more attention.

So overall, I am undecided: I find the story really exciting, but I am convinced that there are too many open questions on the experimental side to let it stand as it is. I have listed these points below (8 major points, and a few minor ones).

As will become clear from my review, I am not an expert in MD simulations, and I recommend that an MD person looks into this closely, too.

Major points

1. One experimental issue is the excitation of the water with a selective pulse (1.5 ms Gaussian pulse). The experiments shown in Figure S2 use this selective pulse as the only element that selects the water-1H polarization. (In fact, the statement on page 3 “The water 1H magnetization was selected by a soft 1H excitation pulse and a T2 filter, then transferred to protein protons during a mixing time tSD.” does not seem to correspond to the pulse sequences in Figure S2, as the latter do not have the T2 filter that the authors refer to. Hence I conclude that it is only the selectivity of the selective pulse that selects the water polarization. And here is also a problem, as the alpha protons have a chemical shift that heavily overlaps with the water frequency. I expect that the experiments measure not only water but also H-alpha properties. And arguably, the final 13C-detected signal may be biased to the signal originally arising from H-alpha, because the H-alpha protons are closer, and thus they are more efficiently transferred to the other aliphatic signals during the spin-diffusion mixing time.

To illustrate this point, I have run a simulation (NMRSIM, within Topspin) of the 1.5 ms Gaussian pulse. Spins within a range of almost 2 kHz (2.5 ppm at 800 MHz) are excited. See the figure in the attached file.

I, thus, think that the experiments are sensitive to both water and H-alpha. The selectivity of their scheme should be tested experimentally with a simple ^1H -detected 1D spectrum, consisting of the Gaussian excitation pulse and acquisition. While the spectrum will of course be dominated by the bulk water signal, I expect to see signals from H-alpha protons close to water. This experiment probably requires a lot of scans, as the bulk water is large compared to the H-alphas and the "channel water". A second experiment could be the type of experiment used here but without spin diffusion. This boils down to a selective excitation with the Gauss pulse followed by a CP and ^{13}C detection. If one can see CA signal, it shall be an indication that the Gaussian pulse excited HA protons. These experiments shall help understand how well the selection works, and how much the HA protons contribute to all the observations used in this manuscript.

2. The contribution of chemical exchange from water to exchangeable sites on the protein: The transfer of ^1H magnetization from water to a protein may contain a significant contribution from chemical exchange combined with spin diffusion. In this mechanism, the proton of water would chemically exchange with a side chain OH, COOH or NH/NH₂, thus bringing the water magnetization into the protein (during the mixing time, denoted as τ_{SD} in this study). Then efficient spin diffusion would occur from this side chain proton to other protons in the protein. This mechanism has been reported to be the dominant mechanism in several instances (see e.g. work by the Zurich group on HETs, JMB 2011). If this is the dominant mechanism, then the apparent buildup, used here extensively to identify water content, would not be due to "proximity to water", as stated here, but actually it would rather reflect the chemical exchange.

Thus, we need to investigate the question whether the observed effects in this study can really be ascribed to proximity of water and protein.

To this end, one shall look into typical exchange rate constants of side chain and backbone protons. The backbone NHs have typically slow exchange, in the many seconds and up to hours/weeks time scales. The exchangeable side chain hydrogen sites, in contrast, often exchange very fast. For acidic side chains the exchange is of the order of 10000 per second (DOI: 10.1021/ja513205s); for arginines it is in the hundreds of milliseconds; for His it is also on the order of 10000-100000 per second, as elegantly shown by Mei Hong's group (e.g. DOI 10.1021/ja2081185). Thus, chemical exchange is clearly very fast, and it certainly occurs during the pulse sequences (e.g. the mixing time τ_{SD}).

How does this impact the interpretation of the results? This should be discussed in the present manuscript, as one might conclude that the interpretation of water buildup spectra in terms of "proximity" is erroneous.

Possibly an argument in favor of the authors' interpretation is the pH dependence. It seems that the buildup in the experimental data is (slightly) faster at lower pH (Figure 2). This observation should be compared to the expected pH dependence of side chain solvent exchange. If intrinsic solvent exchange is faster at lower pH, then the observed effects could be entirely attributed to solvent exchange kinetics, rather than amount of water present, thus questioning the main conclusions of the paper.

The authors recognize the fact that chemical exchange may impact the observed parameters, e.g. (page 4): "These relaxation rates were measured at a sample temperature of 273 K to minimize proton exchange between water and labile protein sites." This of course immediately raises the

question whether lowering the temperature to 273 K would really make a big difference for chemical exchange that is as fast as 10000-100000 per second, and whether cooling to 273 K make make solvent exchange negligible compared to spin diffusion (which occurs at best on a millisecond time scale, i.e. at least 10 to 100 times slower). The authors also recognize the importance of chemical exchange when it comes to water-lipid interaction (page 5: "The rapid relaxation of lipid-associated water is consistent with chemical exchange of water with lipid headgroups.") However, for the analysis of the data in Figure 2, chemical exchange has completely disappeared from the picture.

Overall, I think that the chemical exchange and its possible impact on these experiments need to be discussed. Basically, how is it possibly that the authors can completely neglect (if I am not mistaken!) that chemical exchange may impact apparent buildup rates, even though chemical exchange is as fast as 10000-100000 per second?

3. Relaxation properties of different "water pools"

It is not clear to me how the authors managed to probe selectively the properties of different fractions of water in their sample, i.e. bulk water, membrane-associated water and water in the channel. First it is not clear to me which signals have been used to generate the plots of Figure S3b. Is this the water line (about 5 ppm)? And the water line actually has two peaks; which one was used? And how could the authors ascribe the first part to "lipid-associated water"? Is this based on any spectroscopic technique, or simply because it is the fast component of the decay? The pulse sequences in Figure S3 do not have selective pulses. Was this done with a T2' filter (as suggested in panel e)? And if so, why is this not represented in the pulse sequence of panel S3c?

4. Table S1 lists the rotational correlation times of water, including bulk water. The value of ca. 0.7 ns appears to me quite long; it is a value that small peptides may have, which are more than an order of magnitude larger. I assume that there are very good literature data about water correlation time constants. Could the authors please compare their values to literature data, to make sure that we are in the correct range here?

5. The study addresses water dynamics by measuring apparent transverse relaxation of protons. Here I see a few issues that need to be addressed:

(i) As eluded to above, if this data was obtained with the experiment of Figure S2a, then I believe that we are possibly seeing a superposition of water and H-alpha relaxation.

(ii) Apparent transverse coherence decay (in the manuscript referred to as "relaxation") in solids is generally dominated by effects other than relaxation, i.e. other than dynamics. Coherent effects, in particular the evolution of magnetization due to incompletely averaged dipolar couplings, often largely dominate the apparent decay. This is true for even heteronuclei, but it is even more true for protons, and especially so when the MAS frequency is rather low, which is the case here (14 kHz). In analyzing their ^1H T2' data with BPP theory, the authors assume that one can neglect coherent contributions to decay, but without actually looking into this question. I am not sure whether we can really interpret these data quantitatively (in fact: I doubt). I think that the apparent decay may well contain contributions from water dynamics, but whether this (sought) dynamics contribution is dominant is not clear from the data. Most likely we are seeing a mixture of relaxation and coherent dipolar dephasing.

(iii) I wonder what are the effects of the narrower channel on water coherence decay. In the high-pH closed state, the (rather rigid) helices are closer to the water protons. This means: more dipolar dephasing (not related to dynamics!) and also more of the stronger dipolar couplings which would lead to relaxation (through dynamics). Thus, in any case, I would expect that the high-pH closed state

would have a faster apparent ^1H T_2' coherence decay. This is exactly what is seen, although the effect is modest. And I expect it independently of potential changes in dynamics.

Taken together, I am not convinced that the theoretical framework for interpreting the water- ^1H (and H-alpha?) T_2' decay analysis is solid, as I see theoretical problems, and hence I wonder if the τ_{rot} correlation times are trustworthy.

In this context, the sentence "Transverse relaxation is driven by molecular tumbling, with slower relaxation indicative of shorter rotational correlation times (τ_{rot})." tells only part of the story, and I would not leave it written as it is. In fact, what is seen here is not necessarily relaxation, but an apparent coherence decay which comprises relaxation and most likely also coherent effects.

6. There are open questions regarding the ^1H $R_1\rho$ dispersion experiments, and the interpretation of the results.

In fact, the evolution of the coherence under a spin-lock, at rather low MAS frequency (14 kHz) is fairly complex. First, contains evolution that is strictly independent of dynamics, i.e. in this context it is to be considered an "artefact", in the sense that it would occur even if the molecules were static. Second, the $R_1\rho$ decay contains parts due to dynamics, and here one can cite fluctuations of the chemical shift (which is the only thing the authors consider), but also fluctuations of the dipolar coupling and CSA. All these contributions to the observed $R_1\rho$ decay – those that are dependent on dynamics and those that are not – depend on the RF field strength.

Among the "coherent evolution" part (written under "First" above) I want to mention in particular the dipolar and CSA evolution – which the manuscript then nicely describes in the context of Figures 5 and S6. Under the present conditions, these effects give rise to very fast coherence decay (plus oscillations) at RF fields somewhere from 7 to 30 kHz. This is precisely the range for which dispersions are reported in Figure 3d. Figure S6b actually shows that enhanced $R_1\rho$ is expected for low RF fields – without any molecular motion or chemical- shift fluctuations!

Among the "incoherent evolution" part ("Second" above), is indeed the expected Bloch- McConnell-type dispersion, but also relaxation effects arising due to dipolar/CSA couplings, and in particular when the RF field is close to the rotary resonance conditions. The latter is sometimes referred to as Nerrd effect. Thus, again in the range from 7 to 30 kHz, I would expect higher $R_1\rho$, while as the RF is further away from the rotary resonance conditions (i.e. at higher RF field), the $R_1\rho$ should decrease. This is exactly what is observed.

Therefore, I wonder how much (if any) of the relaxation dispersion can truly be ascribed to isotropic chemical-shift fluctuations, given that one can identify several effects that may produce very similar effects, some of which are even independent of dynamics. What is the argument to claim that all other effects than are negligible?

I understand that one possible argument could be that the bulk water does not show dispersion. But this is not a valid argument, because in the bulk water (freely tumbling), the dipolar/CSA interactions are clearly averaged to zero, which eliminates all the "coherent evolution part" above, as well as the "Nerrd" part above.

I also have problems understanding how the data from Figures 3d and Figure 5a go together. The data in these two experiments have been collected with the exact same pulse sequence, to my understanding. Figure 5 shows that at the rotary resonance conditions (14 kHz, 28 kHz) the signal decays very rapidly, and oscillates. This is expected for the presence of coherent dephasing (my "coherent evolution" part above). If these curves were fitted exponentially, I would estimate the rate constant to be ca. 1-2 ms, i.e. a rate constant of ca. 500 – 1000 s^{-1} . This is very far from the values

reported in Figure 3d. How come? Is this just because the exact rotary resonance conditions were left out? I doubt, as the RF inhomogeneity will make it such that fast decay will occur even 1-2 kHz away (-> reported in Figure S6d).

Let's assume for a moment that everything else than isotropic chemical-shift fluctuations can really be neglected. I wonder if the chemical-shift difference would make sense. If the exchange is really between the two water pools, as claimed, then I expect the chemical-shift difference to be very small (0.1-0.3 ppm), rather than the claimed >2 ppm. In other words, the dispersions are too large for what I think is a realistic chemical-shift difference. Thus, the pretty large dispersions point to something in addition to (or in place of) the isotropic chemical-shift fluctuations. So maybe my hypothesis that the dispersions contain a substantial part of coherent dephasing and/or NMR effects could hold true.

7. The recoupling experiments also have their own challenges.

These experiments are very sensitive to the exact setting of the RF field, as illustrated in Figure S6d. The differences reported in the order parameters (Figure 5a) could easily be "obtained" by mis-setting the RF field strength by 1-2 kHz (I assume). Given that these two experiments have been done with different samples, and thus necessarily with different RF calibrations/tuning etc., it is not trivial to be sure that the RF field was properly set in each sample.

Furthermore, in theory the CSA parameters obtained from the $n=1$ and $n=2$ rotary resonance conditions should be identical, but they are not: in the low-pH sample one finds 2.0 ppm and 1.6 ppm for the two experiments, and for the high-pH sample it is 1.6 ppm and 1.2 ppm. The mis-match between the CSA parameters obtained for a given sample is as large as the difference between the sample. The former is not discussed, but the latter is interpreted in terms of molecular mechanisms.

I was also curious why the data of the spin-lock experiments presented in Figure 5a and 5b are not identical. I assume that the data shown in Figure 5b are from a repeat experiment, but otherwise identical to those shown in Figure 5a. I would claim that the data are not very well reproducible. What would be the fitted CSA parameters from the data shown in Figure 5b?

8. The paper reports that in the low-pH state (open state) water reorients more rapidly. This is based on the reorientational correlation time that comes from ^1H T_2' measurements (Table S1), with all its theoretical problems (see my point 5). At the same time, the oscillations seen in ^1H spin-lock experiments (pulse sequence: Figure S2b; data: Figure 5a middle and right panels). I find it difficult to see how water ^1H can have slower T_2' relaxation (which the authors translate to faster reorientation using BPP theory) and at the same time have higher order parameters. To me this sounds like a contradiction. In the Discussion section, the authors address this as follows:

"The surprising finding of faster water dynamics and higher water anisotropy can be rationalized by the fact that water needs to reorient to break and form hydrogen bonds with adjacent water molecules to permit Grotthuss hopping."

But I do not see how Grotthuss hopping enters the picture here. Grotthuss hopping means that a rearrangement of hydrogen bonding/covalent bonding looks like an effective translation of molecules, although in reality it is not, in the sense that a given atom is actually not moving. NMR is not sensitive to translational motion, only to rotations and fluctuations of inter-atomic distances. Hence, NMR can only see actual movement of motions, but not a "pseudo- movement" as in the Grotthuss mechanism.

Thus, for me the apparent contradiction of shorter T_2' (Fig 3b) and lower order parameter (Fig 5a) in the high-pH state is not resolved. As pointed out in 5. and 7. above, I see several possibilities why the experiments and their interpretation may go wrong, but I don't see how to reconcile the data. Could the authors please explain their view?

By the way, isn't the Grotthuss mechanism impacted by pH? Thus, does changing the pH by 3 units (1000-fold more H+) impact the argument that the authors make here?

Minor comments:

Page 1, bottom: "our recently determined 1.5 Å resolution structures of BM2 in lipid bilayers". The term relating to a certain resolution (here: 1.5 Å) is so closely linked to crystallography, in the minds of almost all structural biologists, that most readers will very likely assume that these are crystal structures. But in fact, they are ssNMR structures. I think this should be made clear. And the bundle RMSD in an NMR structure calculation is not quite the same as the resolution that crystallographers use. I think some clarification would help.

Typo in the SI text, first page: "approximatiely"

The axis in Figure 2d is a bit odd. On the left (and inside the panel) are residue names. However, I see only 8 residue names, but 9 (black) data points; the data points are not really aligned with the ticks next to the residue names. Not clear who is who in this plot.

The caption of Figure S6 states: "Experiments began with x-magnetization and the amount of magnetization remaining on spin #1 was monitored as a function of spin-lock time." The reference to spin #1 is not useful, because the spins are not numbered in the figure. Are the four water molecules symmetry related, i.e. they all have the same environment?

Figure S6d: What do the simulations look like at RF fields above 28 kHz? Are they symmetrical, i.e. is the behavior at 29 kHz identical to the one at 27 kHz?

Figure S6 (simulations of recoupling curves): what is the rationale for the chosen RF field distribution? This is not a realistic distribution for an NMR probe, in my view. It would better be approximated by a Gaussian distribution of RF fields, whereby the tail on the lower RF field side is longer than on the larger RF-field side. Arguably the details of this distribution change very little for the outcome of the fit.

Figure S6b: the simulations are shown for the (unrealistic) case that there is no RF inhomogeneity. For illustration purposes that's ok, but it should be stated explicitly that in a realistic scenario the conditions are broader.

MS: Water Orientation and Dynamics in the Closed and Open Influenza B Virus M2 Proton Channels

Responses to reviews

Reviewer #1 (Remarks to the Author):

A. Summary of the key results

The authors made use of solid-state NMR spectroscopy and molecular dynamics (MD) simulations to characterize the behavior of water in the influenza B M2 protein channel in the low-pH open channel and in the high-pH closed channel.

[1] The number of water molecules contained in the low-pH open channel is approximately the double of that contained in the high-pH closed channel.

[2] Water molecule exhibits rotational motion, which is faster in the low-pH open channel, and subdiffusive dynamics, which is accentuated in the high-pH closed channel. The water flux through the pore is different between the structures, being greater in the low-pH open channel.

[3] H19 and H27 histidine residues cause water to exhibit preferred orientation, with H19 playing a major role in control the orientational order.

B. Originality and significance: if not novel, please include reference.

The study is novel in the sense that the authors report extensive experimental and computational work on influenza B M2 pore channel embedded in a POPE membrane. However, two previous computational papers cited by the authors focus on the role of specific residues as histidines and serines for the channel functionality (see papers by Q.-C Zheng and Q. Zheng cited by the authors, Ref 47 and 48).

The experimental work is remarkable; the computational work has serious shortcomings, and I would not include it -see all my reservations in what follows.

We thank the reviewer for the detailed and thoughtful feedback. We feel the suggested changes have greatly improved the computational work described in the manuscript. To address the reviewer's concerns, the following major improvements have been made to the manuscript: 1) to address the concerns regarding simulations at 277 K we now include control simulations performed at 297 K. The new data, summarized in **Fig. S10**, shows that all of our conclusions (listed by the reviewer in Part A) hold at 297 K. 2) The method section has been overhauled and expanded, to add additional details and citations that will enable the reader to reproduce our simulations. In the following, we indicate the exact changes that were made to address specific feedback.

C. Data & methodology: validity of approach, quality of data, quality of presentation

The authors do not justify the choice of using 1-palmitoyl-2-oleoyl-sn-glycero-3-phosphoethanolamine (POPE) lipid molecules for building the lipid bilayer. This type of lipid molecule is also used for their experimental work but an explanation for this choice is missing. Please, specify the software used to build the lipid bilayer (maybe CHARMM-GUI?).

In “Mandala...Hong, *Nat. Struct. Mol. Biol.*, **2020**, DOI: 10.1038/s41594-019-0371-2” we found that the ^{13}C and ^{15}N chemical shifts were nearly identical for transmembrane residues in POPE, POPC : POPG, and DLPE membranes. Therefore, we chose to use POPE membranes for all experiments and simulations in this study to match up with the membrane that the BM2 structure was solved in. We clarified in the main text that POPE lipid was used in the simulations to match the experimental conditions and added an explicit statement on the choice of membrane for this study in the Methods section

“Nearly identical ^{13}C and ^{15}N chemical shifts were observed for BM2 transmembrane residues in POPE, 1-palmitoyl-2-oleoyl-sn-glycero-3-phosphocholine:1-palmitoyl-2-oleoyl-sn-glycero-3-phospho-(1'-rac-glycerol) POPC:POPG, and (1,2-dilauroyl-sn-glycero-3-phosphoethanolamine) DLPE membranes²⁷. Since the bulk of the structure determination of BM2 in lipid bilayers was done with POPE lipid membranes, the two samples studied herein relied on this membrane”.

A statement was also added in the main text to refer the reader to the Method section where the system setup (including the solvation in POPE lipid) is explained in detail. The Method section “System setup and molecular dynamics simulation protocol” now starts with a statement that CHARMM-GUI was used for system setup.

The statement “The bilayer was solvated by a 30 Å thick water layer on each side of the bilayer with 150 mM NaCl.” is repeated twice (line 3 and 6 of System setup and molecular dynamics simulation protocol paragraph).

As part of an overhaul of the MD method section, based on the reviewer’s comments, this redundant sentence has been removed.

Please specify the force field used for water – being the study focused on water, it is one of the most important pieces of information, if the calculations need to be reproduced. Also, in this work the choice of using an appropriate water model is crucial because the simulations are carried out at the maximum density temperature (TMD), below room temperature (see C. Vega and J. L. F. Abascal, *Phys. Chem. Chem. Phys.*, 2011, 13, 19663–19688).

We now specify in the method section that the TIP3P water model is used in the simulations. We added a note in the methods section regarding potential issues with using TIP3P below room temperature, with the relevant citation. Furthermore, control simulations at 297 K (**Fig. S10**) were added to address the valid concern regarding simulations at 277 K. All of our original conclusions hold for the simulations at 297 K.

Concerning the same topic, it would be important to report a reference of protein simulations carried out with the same force field at low temperature.

We have now added in citations to two peptide/protein MD studies that utilized simulations conducted at 277 K and 278 K, respectively, using the TIP3P water model (Pang, *Biochem. Biophys. Res. Commun.*, 2014, DOI: 10.1016/j.bbrc.2014.08.119; Best & Mittal, *Proteins*, 2011, DOI: 10.1002/prot.22972).

According to the protein data bank website, the (6PVR) closed and (6PVT) open BM2 structures were determined experimentally at 285 K. Please, justify the choice of carrying out a very short equilibration (375 ps in total) at 297 K without performing a heating procedure and the choice of carrying out simulations at 277 K without performing a descendent temperature ramp procedure. Being the simulations carried out at low temperature, the system would need a long equilibration procedure.

We agree with the reviewer that 375 ps of equilibration would be insufficient. In addition to the short 375 ps equilibration protocol, the first 30 ns of each unconstrained NPT simulation was used as additional equilibration time and was excluded from analysis. We have clarified our equilibration protocol in the Method section. We found that after 30 ns of equilibration the system has converged; RMSD (**Fig. S1c**), temperature (277.01 ± 1.35 K to 277.01 ± 1.34 K), volume (578.46 ± 1.25 nm³ to 577.14 ± 1.29 nm³), box-dimensions ([X, Y, Z] = [7.35 ± 0.06 nm to 7.22 ± 0.05 nm, 7.35 ± 0.06 nm to 7.22 ± 0.05 nm, 10.70 ± 0.16 nm to 11.07 ± 0.16 nm]), and potential energy ($-5.59 \cdot 10^5 \pm 1.08 \cdot 10^3$ kJ to $-5.6 \cdot 10^5 \pm 1.14 \cdot 10^3$ kJ), where the quantities in brackets report average \pm standard deviation, separately for simulation time windows between 30-80 ns and 80-130 ns. It should also be noted that all simulations were performed 4-fold (7-fold, when including the new simulations at 297 K), and that all replicates yield consistent results.

The number of steps used for the steepest descent energy-minimization protocol should be written in the main text or there should be a reference for the Supplementary Information. These details of the simulation setup procedure are missing in the text and are important in case the calculations need to be reproduced: the simulation time step, the saving trajectory time step, the values of the harmonic position restraints applied to the protein backbone, protein sidechains, lipid headgroups, and lipid tails. Also, are periodic boundary conditions applied? Is a switching function with a cut off used for the short-range interactions? Which is the time step used for integrating the equations of motion? The simulation time step is important also in terms of the simulation time resolution for computing water autocorrelation functions.

We have overhauled and expanded the MD method section in the main text to include these details. Some of the requested values were already included in **Table S3**; we have now more clearly referenced this table in the main text. To briefly answer the reviewer's questions here: force constants used for the harmonic restraints are listed in **Table S3**, we use periodic boundary conditions, VDW interactions are cut-off at 1.2 nm with a potential shift, we use a 2 fs timestep (enabled via LINCS constraints on h-bonds), and snapshots were stored every 10 ps.

Please, specify how bonds involving hydrogen atoms are treated during the simulation, i.e. if the SHAKE or any other algorithm is used, and how electrostatic interactions are calculated (supposedly using smooth particle-mesh Ewald summation method). Given the importance of the electrostatics and the hydrogens in proton transport these details of the simulation setup are relevant.

These details were added to the expanded method section. Briefly; we used LINCS to treat bonds involving hydrogen atoms, and smooth particle-mesh Ewald summation for electrostatics.

The authors state in the System setup and molecular dynamics simulation protocol paragraph that ‘While most water analysis was conducted on unrestrained simulations, additional simulations were carried out where the protein structure was restrained to the solid-state NMR backbone structure using harmonic restraints’ without describing which structure (open or closed) is being restrained and the values of the harmonic restraints. Some information is written in the legend of Fig S1d, but all information should be reported in the Methods clearly.

These details were added to the method section. Briefly; we used $50 \text{ kJ mol}^{-1} \text{ nm}^{-2}$ harmonic restraints on the backbone atoms to keep the BM2 structure near the resolved NMR structure (PDB: 6PVT for open +4/+4 H19/H27, and PDB: 6PVR for closed 0/+1 H19/H27) for control simulations (**Fig. S1, S4**), or to keep the BM2 structure near the resolved NMR structure while switching the H19/H27 charge state to investigate the effect of the H19/H27 charge state on water-orientation in the absence of large-scale conformational changes (**Fig. 5f, S9**) (PDB: 6PVT for open 0/+1 H19/H27, and PDB: 6PVR for closed +4/+4 H19/H27).

The authors provide a movie without the reference to it in the main text.

References to the movie were added in the main text.

The authors state: ‘The equilibrated structures were next subjected to four independent production runs of 130 ns in an NPT ensemble, with semi-isotropic pressure coupling at 1 atm and 277 K.’ Please, specify the algorithms used to maintain the pressure and temperature constant and the reference to the corresponding paper if any.

We used the Parrinello-Rahman barostat and the V-rescale thermostat. These details are now added to the methods section.

Please specify whether the average reported in table S4 for assigning the values of the channel-axis coordinates Z is calculated from the initial structure atom coordinate or from the equilibrated structure.

The reported averages are over the production simulations. This is now clarified in the **Table S4** legend.

Data reported in Fig S4a for the unrestrained simulations should be reported in scattered format not with a dashed line and with their error bars for allowing proper comparison with the restrained simulation data.

For visual clarity, we used dashed lines to re-plot the main text data as a comparison to the restrained data in **Fig. S4**. Data from **Fig. 2e** appears in **Fig. S4a (left)**; data from **Fig. 3f** appears in **Fig. S4a (middle)**; data from **Fig. 5e** appears in **Fig. S4a (right)**. A plot with two different scatter points and error bars (*vide infra*) looks too cluttered. The point of **Fig S4** is to show that none of the conclusions are sensitive to the structural changes in BM2 that were observed in the

unconstrained MD simulations. In our opinion, **Fig S4** as currently plotted successfully conveys this point, and the error bar data is already available in the main text.

An alternative version of **Fig S4**, with scatter points and error bars is included here for your reference.

The authors define accurately and correctly how they quantify the water translational dynamics and the water flux through the channels in the Methods. However, they are inaccurate when they describe their results in the main text (p.7 'We also examined water translational diffusion in the simulations, by tracking the number of water molecules that passed through the channel in either direction' and 'We next computed the mean-squared displacement (MSD) of water molecules in the channel (Fig. S4b),...'). Authors should maintain the definition of flux in the main text or consider using a collective diffusion model (see for instance F. Zhu, E. Tajkhorshid, and K. Schulten, PRL 93, 224501, 2004) if they discuss the translational diffusion through pores in terms of water permeation through channels.

We have reworded the following sentence in the main text to define more accurately and consistently how water transport across the channel was quantified. See below:

“We also examined the flux of water molecules through the channel, by counting the number of water molecules that entered the channel on one side and exited at the other side (counted as one transport event).”

The authors state: ‘The number of water molecules increases the most near the hydrophobic valve formed by L8,...’, which seems odd given the hydrophobic character of leucine residues. This could be due to the dimension/radius of the pore or to the pore electrostatics. It would be interesting to investigate further.

Based on the reviewer’s comment, we now more directly relate the increased amount of water near L8 to the channel diameter (**Fig. S1b**) in the Discussion:

*“The lower water amount at L8 in the closed^{0/+1} BM2 channel is therefore directly correlated with a more compact N-terminal pore. The minimal diagonal heavy-atom distance at L8 is 11 Å for the open^{+4/+4} channel and 5.5 Å for the closed^{0/+1} channel (**Fig. S1b**).”*

It is important to note that although the amount of water near L8 is most increased in the open channel relative to the closed channel (**Fig. 2e**), the total amount of water near L8 is still quite low (**Fig. 2d**).

In Figure 2d the data at the channel-axis coordinate Z values of approximately -20, 15, and 20 are missing, whereas the relative ratio between those values is reported in Fig 2e. The plot in Fig 2d should contain all data or be displayed with all data in the Supplementary Information or the author should justify their choice of not showing such data.

This data was not shown in panel D because, due to the channel widening at the exits, there is much more water at those Z-values, which are outside the x-axis range of 0–30 for $\langle N_{\text{water}} \rangle$. Moreover, there is not much difference in water amounts between the open and closed channel (**Fig. 2e**). We chose a range of 0 – 30 for the $\langle N_{\text{water}} \rangle$ axis in order to highlight the difference inside the channel. We have now added a justification for the x-axis value in the figure caption.

“To focus on the differences in the channel interior, data points for which $\langle N_{\text{water}} \rangle$ is greater than 30 are not shown.”

In Fig 5e,f the authors report the results of the analysis of the water orientational order performed on the unrestrained and restrained simulations. The fact that by inverting the histidine charges between the closed and open structures, the original orientation of the dipoles is not recover completely (closed^{0/+1} differs from open^{0/+1} structure) indicates that there are more factors than the histidine charges influencing the dipole direction. These factors could be the electrostatic potential of the whole channel or system, or the geometry of the channel. Without exploring these factors, it is hard to demonstrate that the dipole alignment depends only on those charges.

Our conclusion only says that the H19 charge state is essential for controlling the water dipole orientation; we do not say that it is the only factor. The simulations demonstrate that one can

achieve the observed water orientation in the closed channel by only changing the H19 charge (compare the black line in **Fig 5e** to the orange line in **Fig 5f**).

From AM2 studies, we know that the protein conformation and conformational dynamics control the directionality and limit the flux of protons. We hypothesize that the same may be true for BM2. This protein conformational effect does not directly address water dipole orientation. We have now revised the text in the discussion to clarify this point:

“For AM2, the protein conformation and dynamics play the roles of limiting the net proton flux and dictating the directionality of the proton flux^{24,25}. Whether proton flux in BM2 is rate-limited by protein conformational dynamics has yet to be shown, but the similar rates of proton conduction and conserved HxxxW motif suggest that this may also be the case for BM2.”

D. Appropriate use of statistics and treatment of uncertainties Appropriate use.

E. Conclusions: robustness, validity, reliability

The authors state: ‘We attribute the fast decay in ~10 ps to thermal motion of bulk-like water, and the slow decay to collective dynamics of the channel-water hydrogen-bonding network in the confined pore and interactions of water with protein residues’. The reference to “collective dynamics” is inaccurate and needs a better explanation. The rotational auto-correlation function as defined in the Methods provides information on the rotational motion of the dipole vector of a water molecule. Then, the function is calculated over all molecules and averaged, but it does not report on collective dynamics, which can be computed differently by means of MD simulations (see for instance S. Capponi, S. H. White, D. J. Tobias, M. Heyden, J. Phys. Chem. B 123 (2), 480-486, 2018; V. C. Nibali, G. D’Angelo, A. Paciaroni, D. J. Tobias, M. Tarek. J. Phys. Chem. Lett., 5, 1181–1186, 2014; M. Tarek and D. J. Tobias, PRL, 89, 275501, 2002).

The reviewer is correct to point out that our current analysis does not inform on “collective dynamics”, merely on the timescale of the water dynamics as a population average. We have removed the phrase “collective dynamics” and restate our result as follows:

“We attribute the fast decay within ~10 ps to thermal motion of bulk-like water, and the slow decay to motion of the channel-water hydrogen-bonded water molecules in the confined pore and interactions of water with protein residues.”

In the same statement about rotational dynamics, the authors wrote: ‘We attribute the fast decay in ~10 ps to thermal motion of bulk-like water’. The characteristic rotational relaxation time of bulk-like water at room temperature is ~1 ps. With decreasing temperature, this characteristic time becomes slower, but without a proper discussion about how temperature affects rotational dynamics (simulations are carried out at 277 K) and about the water force field it is difficult to assign the fast decay of ~10 ps to thermal motion of bulk-like water. Also, at a given temperature water in confined geometries shows slower dynamics compared to bulk-like water (Teixeira, J., Zanotti, J.-M., Bellissent-Funel, M.-C., Chen, S.-H. Phys. B 1997, 234–236, 370; Bellissent-Funel, M.-C. J. Phys.: Condens. Matter 2001, 13, 9165; Starr, F. W., Nielsen, J. K., Stanley, H.

E. Phys. Rev. Lett. 1999, 82, 2294.). Authors should comment on that because water in BM2 is confined.

Due to the time-resolution of our simulation snapshots (10 ps) we cannot further resolve the fast rotational dynamics, we only know there are dynamics occurring that are faster than 10 ps. In order to directly compare to the timescales investigated in the NMR experiments, we have focused on the long-timescale rotations in this work. We have added a comment to the text that we cannot further resolve the dynamics faster than 10 ps, and that these dynamics could also be affected by the confinement within the channel (citing the appropriate references as listed by the reviewer).

“Due to the time resolution used in the analysis of our MD simulations, we cannot further comment on the sub-10 ps rotational time, although it is possible that this motion is also affected by confinement within the channel 47,48.”

As noted by the reviewer, temperature is also a factor, though this is now addressed by the newly added simulations at 297 K (see **Fig. S10**).

Please provide a reference or an explanation about the electrostatic repulsion on this statement: ‘... the four-helix bundle is more loosely packed at low pH due to electrostatic repulsion of the +4 H19 tetrad.’

We have reworded the sentence to clarify that the electrostatic repulsion referred to here is between the four positively charged sidechains that make up the H19 tetrad:

“...the four-helix bundle is more loosely packed at low pH due to electrostatic repulsion between the four positively charged H19 residues.”

We have also added in citations to papers that show an expanded, or more loosely packed, pore at low pH.

This statement is unclear and it should be discussed appropriately given the temperature at which the simulations are carried out: ‘Further, this agreement suggests that the difference in water occupancy in the channel is insensitive to the effects of thermal fluctuations’

This is addressed by the new set of simulations at 297 K, the number of water molecules in the open channel is consistently higher than in the closed channel. We have now updated the statement in the discussion to include the new data:

“Further, the increased amount of water is not an artefact of the low temperature (277 K) at which the experiments were conducted, as shown by control simulations at 297 K (Fig. S10).”

The authors state: ‘The lower water amount at L8 in the closed^{0/+1} BM2 channel is therefore directly correlated with a compact N-terminal pore.’ It would be useful to have quantitative data or a reference to describe what “compact” means, as for instance the dimension of the radius of the pore at L8 site in the closed and open structure.

We have added a quantitative comparison of the pore diameter at L8 in the Discussion, and a reference to **Figure S1b** which shows the pore diameter throughout the channel.

“The lower water amount at L8 in the closed^{0/+1} BM2 channel is therefore directly correlated with a more compact N-terminal pore. The minimal diagonal heavy-atom distance at L8 is 11 Å for the open^{+4/+4} channel and 5.5 Å for the closed^{0/+1} channel (Fig. S1b).”

In addition, we have quantified the channel compaction using the quadrilateral area between the four helices (by calculating the areas spanned by the four centers of mass of side chain heavy atoms on each of the tetramer helices for particular residues- whether all or some side chain heavy atoms were included in center of mass determination depended on properties of individual residues). The result of this analysis is highly similar to **Fig. S1b**.

The authors state: ‘Therefore, the polarity of the hydrogen-bond network is electrostatically controlled by the charge state of the histidine. In comparison, the protein conformation and dynamics play the roles of limiting the net proton flux and dictating the directionality of the proton flux.’ Further analysis on the conformation and shape of the pore and on the dynamics of the protein, i. e. of the residues and side chains lining the pore, should be carried out to support this statement.

MD studies of AM2 have predicted that protein conformational dynamics limit the flux of protons through the channel (Acharya.. .Klein, A. *Proc. Natl Acad. Sci. USA*, **2010**, DOI: 10.1073/pnas.1007071107). A later SSNMR study showed that AM2 protein conformational exchange happens on the same timescale as proton conduction Mandala.. .Hong, *J. Am. Chem. Soc.*, **2018**, DOI: 10.1021/jacs.7b12464), providing experimental evidence in support of the conclusions from the earlier MD study. However, we do not have evidence for this being the case for BM2 and it is beyond the scope of this study. We have thus reworded and clarified this section as such,

“For AM2, the protein conformation and dynamics play the roles of limiting the net proton flux and dictating the directionality of the proton flux^{24,25}. Whether proton flux in BM2 is limited by protein conformational dynamics has yet to be shown, but the similar rates of proton conduction and conserved HxxxW motif suggest that this may also be the case for BM2.”

The author state: ‘Our results show that proton transport by BM2 requires fast water rotational and translational diffusion that optimizes the orientation of a hydrogen-bonded water wire.’ In order to make this conclusion plain and comprehensible, the authors should provide details or references to explain how the rotational and translational diffusion of water are related to the proton transport and how the results presented in this work fit into the big picture.

To better distinguish water properties from proton transport behavior, we have now revised the sentence to reflect the correlation, rather than causation, between the two:

“Our results show that proton transport by BM2 in its open state is correlated with fast water rotational and translational diffusion, which optimizes the orientation of a hydrogen-bonded water wire.”

We have also added citations to two studies (Laage, D. and Hynes, J.T. *Science*. 311 (5762), 832-835 (2006) DOI: 10.1126/science.1122154; Hassanali, A. et al. *Proc. Natl. Acad. Sci. USA*, 110 (34) 13723-13728 (2013) DOI: [10.1073/pnas.1306642110](https://doi.org/10.1073/pnas.1306642110)) that support large-amplitude rotational and translational motion of water molecules being important for proton transfer.

F. Suggested improvements: experiments, data for possible revision

Would it be possible to provide a figure representing the low-pH and high-pH BM2 structures overlapping? It would help the reader to understand better the differences between the open and closed structure at the beginning and at the end of the simulation and the size and geometry of the cavity without looking at the structure paper (Mandala, V.S., Loftis, A.R., Shcherbakov, A.A., Pentelute, B.L. & Hong, M. *Nat. Struct. Mol. Biol.* 27, 160-167 (2020)).

We have now added a new panel in **Fig. S1a** that compares the SSNMR structures of the open low-pH channel and closed high-pH channel. An additional comparison between the two channels can be found in Figure 1, which is more focused on the residues lining the pore and the pore water molecules.

The authors discuss extensively how temperature affects the experiments carried out on the AM2 channels. A discussion on how temperature might affect the experiments on the BM2 channels should be provided as well. Given the variability of the performance of water model and the computational complexity of describing the interactions between water and protein at low temperature with such models, it would be useful to perform simulations at room temperature and discuss the results.

We have now performed MD simulations at 297 K, partly as a control to alleviate concerns related to simulations at 277 K, and also to discuss the effect of this change in the temperature on the results. These data are summarized and discussed in **Fig. S10**. We also note the simulations and the main conclusions in the methods and discussion sections.

G. References: appropriate credit to previous work?

References to computational work related to the setup of the simulations should be included along with adding the missing details which I underlined in the previous comments (part A). The rest of the work gives appropriate credit to previous study.

Done as requested.

H. Clarity and context: lucidity of abstract/summary, appropriateness of abstract, introduction and conclusions

The abstract and the introduction are clearly presented. The authors should spend few more words in the introduction on the difference between influenza A M2 protein and influenza B M2 protein to guide properly the reader unfamiliar with these systems.

We have now added the following two sentences to the introduction:

“This uniform expansion of the BM2 pore at low pH correlates with the ability of BM2 to conduct protons bidirectionally, down the concentration gradient, like a canonical ion channel. In comparison, AM2’s TM domain exhibits a helical kink at G34, which acts as a hinge to alternate water access to the N- or C-terminal halves of the pore^{11,24}. This distinct conformational motion correlates with AM2’s function to conduct protons strictly inward, like a transporter.”

For a more extensive comparison of the similarities and differences between AM2 and BM2, we refer the readers to the publication Mandala...Hong, *Nat. Struct. Mol. Biol.*, **2020**, DOI: 10.1038/s41594-019-0371-2.

Note: in addition to the changes made to address the reviewer concerns, we have also updated Figure panels **5f**, **S1d**, **S9a**, and **S9b**. In the original submission, data from simulations of the “+4/+1 pH 7.5 structure” was shown, while the figure labels state “+4/+4 pH 7.5 structure”. The updated figure panels use the correct “+4/+4 pH 7.5 structure” simulation data. Since the only difference in these simulations is the protonation state of H27, not the critical H19, the changes to the figure are minor and the manuscript text is unaffected.

Reviewer #2

This is a highly interesting manuscript. It describes the properties of water in an ion channel (M2) in two different states, an open state obtained at low pH, and a closed state obtained at high (neutral) pH. This work build upon beautiful work by the Hong group on M2, including high-resolution structures of these two states (NSMB, Mandala et al 2020). It combines solid-state NMR with MD simulations and shows, for example, that

- the open state contains about 2-fold more water than the closed one
- water has different dynamics in the two states, leading to different (>30-fold) water transport
- there are “bottleneck” positions along the channel for water transport

The study used a number of ssNMR techniques to decipher the properties of water. In many cases, the story appears coherent, and there are quantitative matches between properties obtained by NMR and those obtained by MD simulations.

I think that this story will interest many researchers in different fields (channels; NMR; hydration dynamics), and thus I recommend that at some point it gets published. Among the elegant points of the paper (also worth being in the public domain) is the derivation of spin diffusion rates in the SI text.

We thank the reviewer for these very positive comments about the current manuscript as well as our lab’s previous work on influenza M2, and for recognizing the broad appeal of this study to a wide scientific audience.

This being said, from an NMR point of view it contains so many open points, where I think that the authors have taken shortcuts that must be investigated again. Briefly, I think that for almost every NMR observable there are either severe experimental pitfalls or even theoretical issues, that I am not sure that the conclusions really derive from the data in an unambiguous manner.

While I don’t think that this is the case – I know Mei Hong’s rigorous work for long time – it might even be that many of the matches are fortuitous. In any case, the interpretation of a number of parameters really needs more attention.

So overall, I am undecided: I find the story really exciting, but I am convinced that there are too many open questions on the experimental side to let it stand as it is. I have listed these points below (8 major points, and a few minor ones).

We thank the reviewer for providing critical and constructive feedback on our experimental execution and interpretation. We have now carried out additional control experiments and analysis to address the specific critiques of the reviewer. We believe that these changes address all of the reviewer’s concerns and should clear up any ambiguities in our interpretation.

As will become clear from my review, I am not an expert in MD simulations, and I recommend that an MD person looks into this closely, too.

Major points

1. One experimental issue is the excitation of the water with a selective pulse (1.5 ms Gaussian pulse). The experiments shown in Figure S2 use this selective pulse as the only element that selects the water-1H polarization. (In fact, the statement on page 3 “The water 1H magnetization was selected by a soft 1H excitation pulse and a T2 filter, then transferred to protein protons during a mixing time tSD.” does not seem to correspond to the pulse sequences in Figure S2, as the latter do not have the T2 filter that the authors refer to. Hence I conclude that it is only the selectivity of the selective pulse that selects the water polarization. And here is also a problem, as the alpha protons have a chemical shift that heavily overlaps with the water frequency. I expect that the experiments measure not only water but also H-alpha properties. And arguably, the final 13C-detected signal may be biased to the signal originally arising from H-alpha, because the H-alpha protons are closer, and thus they are more efficiently transferred to the other aliphatic signals during the spin-diffusion mixing time.

We apologize for the confusion here. The hydration data shown in Fig. 2 utilized the same pulse sequence as the 13C-detected 1H T2 experiment (Fig. S2a) and has a built-in echo time of $0.143 \text{ ms} \times 2 = 0.286 \text{ ms}$, which is equal to a total of 4 rotor periods under 14 kHz MAS. We have now clarified in the text the length of this T2 filter.

To verify whether this 1H T2 filter length of 0.286 ms, together with the 1H selective excitation pulse, are sufficient to suppress the H α proton magnetization, we ran a control experiment with a short 1H spin diffusion time of 0.1 ms and detected the 13C spectra. The figure below shows that no 13C signals survive under these conditions, confirming that the peptide 1H T2's at 14 kHz MAS are much shorter than 0.286 ms.

The reviewer is correct that our original 13C-detected 1H T1 ρ relaxation experiment did not include the 1H T2 filter. Thus, to verify the original conclusions, we have carried out additional control experiments with short spin diffusion times to check that no protein 1H magnetization remains (*vide infra*).

To illustrate this point, I have run a simulation (NMRSIM, within Topspin) of the 1.5 ms Gaussian pulse. Spins within a range of almost 2 kHz (2.5 ppm at 800 MHz) are excited. See figure below.

I, thus, think that the experiments are sensitive to both water and H-alpha. The selectivity of their scheme should be tested experimentally with a simple ^1H -detected 1D spectrum, consisting of the Gaussian excitation pulse and acquisition. While the spectrum will of course be dominated by the bulk water signal, I expect to see signals from H-alpha protons close to water. This experiment probably requires a lot of scans, as the bulk water is large compared to the H-alphas and the “channel water”. A second experiment could be the type of experiment used here but without spin diffusion. This boils down to a selective excitation with the Gauss pulse followed by a CP and ^{13}C detection. If one can see CA signal, it shall be an indication that the Gaussian pulse excited HA protons. These experiments shall help understand how well the selection works, and how much the HA protons contribute to all the observations used in this manuscript.

We agree that it is crucial to verify that no $\text{C}\alpha$ polarization is detected in the water-edited experiments or the ^{13}C -detected ^1H relaxation experiments. To do this, we carry out a control 1D experiment using a vanishingly short ^1H spin-diffusion period to check that no detectable protein ^{13}C signal is observed. This spin diffusion time is 0.1 ms for ^{13}C -detected ^1H T_2 measurements and 1 μs for the ^1H $T_{1\rho}$ and ^1H spin-lock recoupling experiments conducted in this study. If no protein ^{13}C signal is detected after these short mixing times, then we are confident that the protein ^1H magnetization is sufficiently removed and the final detected ^{13}C signal originates from water protons. We have added a description of these control experiments to the Methods section.

Below we plot the ^{13}C aliphatic region of the control experiments and compare them to spectra measured with longer spin diffusion times and to regular ^{13}C CP spectra. Indeed, the 1 μs control

spectra do not show any ^{13}C signals. Thus, there is no appreciable contribution from $\text{H}\alpha$ protons. We also re-measured the spin-lock recoupling spectra after inserting a 1 ms ^1H T_2 filter and did not observe a noticeable difference. This confirms that the reported ^1H anisotropy indeed comes from water protons.

2. The contribution of chemical exchange from water to exchangeable sites on the protein: The transfer of ^1H magnetization from water to a protein may contain a significant contribution from chemical exchange combined with spin diffusion. In this mechanism, the proton of water would chemically exchange with a side chain OH, COOH or NH/NH₂, thus bringing the water magnetization into the protein (during the mixing time, denoted as τ_{SD} in this study). Then efficient spin diffusion would occur from this side chain proton to other protons in the protein. This mechanism has been reported to be the dominant mechanism in several instances (see e.g. work by the Zurich group on HETs, JMB 2011). If this is the dominant mechanism, then the apparent buildup, used here extensively to identify water content, would not be due to “proximity to water”, as stated here, but actually it would rather reflect the chemical exchange.

We agree that chemical exchange coexists with spin diffusion as two mechanisms for the observed polarization transfer buildup curves. We have now updated the text to indicate that proton polarization transfer is a mixture of spin-diffusion and chemical exchange and have

changed τ_{SD} to τ_{mix} . The work on HETs by the ETH group used a perdeuterated protein under fast MAS (55 kHz) to minimize spin-diffusion effects. Therefore, we cannot use that study to rule out a combined chemical exchange and spin-diffusion mechanism.

Thus, we need to investigate the question whether the observed effects in this study can really be ascribed to proximity of water and protein.

To this end, one shall look into typical exchange rate constants of side chain and backbone protons. The backbone NHs have typically slow exchange, in the many seconds and up to hours/weeks timescales. The exchangeable side chain hydrogen sites, in contrast, often exchange very fast. For acidic side chains the exchange is of the order of 10000 per second (DOI: 10.1021/ja513205s); for arginines it is in the hundreds of milliseconds; for His it is also on the order of 10000-100000 per second, as elegantly shown by Mei Hong's group (e.g. DOI 10.1021/ja2081185). Thus, chemical exchange is clearly very fast, and it certainly occurs during the pulse sequences (e.g. the mixing time τ_{SD}).

Backbone NH exchange rates are very slow for residues involved in hydrogen-bonded secondary structures. Skelton et al. and Linse et al. reported exchange rates of 10^{-1} to 10^{-7} s^{-1} at 300 K for these types of protons at pH 6.0 and pH 7.25. Since our experiments were conducted at ~25 K colder temperature, we expect even slower exchange. Even exposed amide protons from random coil peptides have slow exchange, particularly at pH 4.5 (Englander...Kallenbach, *Quart. Rev. Biophys.* **1984**, DOI: [10.1017/S0033583500005217](https://doi.org/10.1017/S0033583500005217)). Therefore, we do not expect chemical exchange from water to backbone amide protons to make any contribution to ^1H polarization transfer during a mixing time of a few milliseconds.

We address sidechain hydrogen exchange rates below.

How does this impact the interpretation of the results? This should be discussed in the present manuscript, as one might conclude that the interpretation of water buildup spectra in terms of "proximity" is erroneous. Possibly an argument in favor of the authors' interpretation is the pH dependence. It seems that the buildup in the experimental data is (slightly) faster at lower pH (Figure 2). This observation should be compared to the expected pH dependence of side chain solvent exchange. If intrinsic solvent exchange is faster at lower pH, then the observed effects could be entirely attributed to solvent exchange kinetics, rather than amount of water present, thus questioning the main conclusions of the paper.

Indeed, as the reviewer points out, the intrinsic proton exchange rates are slower at pH 4.5 than at pH 7.5, thus strengthening the conclusion that the low-pH channel, which exhibits faster water-protein polarization transfer, contains more water. Specifically, histidine sidechain H_N chemical exchange has rates of 10^4 s^{-1} at pH 4.5 and 10^5 s^{-1} at pH 7.5 (Hu...Hong, *J. Am. Chem. Soc.*, **2012**, DOI: 10.1021/ja2081185; Sehgal...Pelupessy, *Chem. Eur. J.*, **2014**, DOI: 10.1002/chem.201304992). Tryptophan sidechain hydrogen exchange rate is on the order of 1 at pH 4.5 to 100 s^{-1} at pH 7.5 at 300 K; we expect the rates to be about an order of magnitude slower at our lower experimental temperature. Serine sidechain proton exchange rate has been reported to be $\sim 300 \text{ s}^{-1}$ and roughly the same at pH 4.5 and pH 7.5 at 277 K (Liepinsch...Wuthrich, *J. Biomol. NMR*, 1992, DOI: doi.org/10.1007/BF02192808). Therefore,

chemical exchange is overall slower for the pH 4.5 sample than the pH 7.5 sample. Thus, our qualitative conclusion that the low pH channel has more water than the high pH channel holds, regardless of how different the effective diffusion coefficients of magnetization transfer are.

We have now added a section on page 4 to spell out the pH-dependence of chemical exchange rates, and to clarify that the measured polarization transfer buildup curves at low and high pH supports the interpretation of the low-pH channel containing more water.

The authors recognize the fact that chemical exchange may impact the observed parameters, e.g. (page 4): “These relaxation rates were measured at a sample temperature of 273 K to minimize proton exchange between water and labile protein sites.” This of course immediately raises the question whether lowering the temperature to 273 K would really make a big difference for chemical exchange that is as fast as 10000-100000 per second, and whether cooling to 273 K make solvent exchange negligible compared to spin diffusion (which occurs at best on a millisecond time scale, i.e. at least 10 to 100 times slower). The authors also recognize the importance of chemical exchange when it comes to water-lipid interaction (page 5: “The rapid relaxation of lipid-associated water is consistent with chemical exchange of water with lipid headgroups.”) However, for the analysis of the data in Figure 2, chemical exchange has completely disappeared from the picture.

We apologize for the un-systematic way of discussing chemical exchange in the data interpretation. We have now revamped our description of the data in Figure 2 on page 4, to include both chemical exchange and spin diffusion from the get-go.

We note that having an experimental observable that includes both hydrogen exchange and ¹H spin diffusion effects does not actually affect the statement that the “water accessibility” of a protein is higher at low pH than at high pH, because for chemical exchange to occur, there must be water molecules next to the labile protein protons.

Overall, I think that the chemical exchange and its possible impact on these experiments need to be discussed. Basically, how is it possibly that the authors can completely neglect (if I am not mistaken!) that chemical exchange may impact apparent buildup rates, even though chemical exchange is as fast as 10000-100000 per second?

We agree that water polarization transfer occurs through multiple mechanisms, including chemical exchange and spin diffusion (See e.g. Ader...Baldus, *J. Am. Chem. Soc.* **2008**, DOI: 10.1021/ja806306e; Luo & Hong, *J. Am. Chem. Soc.* **2010**, DOI: 10.1021/ja9096219; Williams & Hong, *J. Magn. Reson.* **2014**, DOI: 10.1016/j.jmr.2014.08.007).

However, the contribution of chemical exchange to polarization transfer can be absorbed into the “diffusion” model outlined in the Supporting Information, with the modification that

“the relative intensities of the closed and open channels in the short- τ_{mix} limit reflect the relative magnitude of the product of the water amount and the square root of the effective diffusion coefficient (SI Text, Eq. 5).”

Thus, the correction is that we call this model a “magnetization transfer” model, which includes both chemical exchange and spin diffusion, rather than a pure “spin diffusion” model.

For our BM2 data shown here, the measured buildup rate ratio of 1.95 represents the lower limit of the ratio of water in the pH 4.5 sample compared to the pH 7.5 sample, to account for the fact that chemical exchange is slower at pH 4.5 compared to pH 7.5. This semi-quantitative statement remains correct.

3. Relaxation properties of different “water pools”

It is not clear to me how the authors managed to probe selectively the properties of different fractions of water in their sample, i.e. bulk water, membrane-associated water and water in the channel. First it is not clear to me which signals have been used to generate the plots of Figure S3b. Is this the water line (about 5 ppm)? And the water line actually has two peaks; which one was used? And how could the authors ascribe the first part to “lipid-associated water”? Is this based on any spectroscopic technique, or simply because it is the fast component of the decay?

We integrated across the entire water signal (between the dashed lines) for our ^1H -detected relaxation measurements. The splitting observed in the water peak is due to the shim being temperature-sensitive and the shimming was done at room temperature rather than the experimental temperature. Re-shimming at the experimental temperature and repeating our relaxation experiments does not change the observed relaxation times.

The ^1H -detected T_2' relaxation curves did not fit to a mono-exponential curve, but fit very well to a biexponential curve. We assume that the dynamic bulk-like water in our sample has the slowest T_2' relaxation. The fast relaxing component contributes a population percentage of 23% and 32% for the pH 7.5 and pH 4.5 samples, respectively. For well-hydrated multilamellar POPC vesicles, each lipid headgroup has 9.4 water molecules tightly bound to it and a corresponding 31.0 interlamellar water molecules associated with it (Kucerka...Nagle, *Membr. Biol.*, **2005**, DOI: 10.1007/s00232-005-7006-8). We assume that this ratio is not qualitatively different for POPE membranes, thus we estimate that ~30% of the water should be tightly bound to lipid headgroups and in fast chemical exchange. This is consistent with the population percentages we have attributed to “lipid-associated water”. Because this population has retarded rotational motion due to its interaction with the lipid headgroups and because it exchanges with the headgroup, we expect its T_2' relaxation to be very fast, which is indeed what we observe.

The pulse sequences in Figure S3 do not have selective pulses. Was this done with a T_2' filter (as suggested in panel e)? And if so, why is this not represented in the pulse sequence of panel S3c?

We did not utilize selective pulses for our ^1H -detected experiments because we integrate across either the water peak or the lipid CH_2 & CH_3 peaks, neither of which have appreciable signal from other sources; with a proton detected experiment we can separate what spins contribute based on the ^1H chemical shift. A T_2' filter was used for the bottom figure on what was previously panel (e) (now panel (f) to select for the mobile (bulk-like) water. However, a T_2' filter was not used for the top plot in what is now (f) or for what is now panel (g). We now explicitly show the T_2' filtered $R_{1\rho}$ pulse sequence in panel (c).

4. Table S1 lists the rotational correlation times of water, including bulk water. The value of ca. 0.7 ns appears to me quite long; it is a value that small peptides may have, which are more than an order of magnitude larger. I assume that there are very good literature data about water correlation time constants. Could the authors please compare their values to literature data, to make sure that we are in the correct range here?

We have now updated the text to always refer to bulk-like water, rather than bulk water. Due to water in our samples being part of hydrated multilamellar lipid vesicles, there is no true bulk water. True bulk water has a ^1H T_2 of ca. 500 ms, while the bulk-like water we observe relaxes an order of magnitude faster. Furthermore, as mentioned below, nuclear interactions such as CSA and dipolar couplings are not completely averaged in our samples, thus our reported correlation times should be taken as an upper limit to the actual rotational correlation times.

Bulk water at 275 K has a rotational correlation time of 5.8 ps (Ropp...Skinner, *J. Am. Chem. Soc.*, **2001**, DOI: 10.1021/ja010312h). We have now cited this paper and added the following to the manuscript:

“The bulk-like water in our samples is far less mobile than truly isotropic pure water, which has a correlation time of 5.8 ps at 275 K.”

5. The study addresses water dynamics by measuring apparent transverse relaxation of protons. Here I see a few issues that need to be addressed:

(i) As eluded to above, if this data was obtained with the experiment of Figure S2a, then I believe that we are possibly seeing a superposition of water and H-alpha relaxation.

H α 's do not contribute significantly to our detected signal (see explanation above to “1”).

(ii) Apparent transverse coherence decay (in the manuscript referred to as “relaxation”) in solids is generally dominated by effects other than relaxation, i.e. other than dynamics. Coherent effects, in particular the evolution of magnetization due to incompletely averaged dipolar couplings, often largely dominate the apparent decay. This is true for even heteronuclei, but it is even more true for protons, and especially so when the MAS frequency is rather low, which is the case here (14 kHz). In analyzing their ^1H T_2' data with BPP theory, the authors assume that one can neglect coherent contributions to decay, but without actually looking into this question. I am not sure whether we can really interpret these data quantitatively (in fact: I doubt). I think that the apparent decay may well contain contributions from water dynamics, but whether this (sought) dynamics contribution is dominant is not clear from the data. Most likely we are seeing a mixture of relaxation and coherent dipolar dephasing.

We agree with the reviewer that our observed T_2' are not completely due to relaxation and that the scenario we are studying does not meet all the assumptions for BPP theory. We applied BPP theory to use the simplest model that can give some qualitative insight and a comparison with the MD results. We have expanded upon the following sentence in the text to clarify that our reported correlation times are likely an overestimate:

“Because the channel-bound water has a small degree of orientational anisotropy (vide infra), coherent effects can accelerate the apparent transverse relaxation compared to that caused by molecular tumbling alone. Nevertheless, we can estimate an upper limit of τ_{rot} from the Bloembergen-Purcell-Pound (BPP) theory³⁸.”

(iii) I wonder what are the effects of the narrower channel on water coherence decay. In the high-pH closed state, the (rather rigid) helices are closer to the water protons. This means: more dipolar dephasing (not related to dynamics!) and also more of the stronger dipolar couplings which would lead to relaxation (through dynamics). Thus, in any case, I would expect that the high-pH closed state would have a faster apparent $1H T_2'$ coherence decay. This is exactly what is seen, although the effect is modest. And I expect it independently of potential changes in dynamics.

Taken together, I am not convinced that the theoretical framework for interpreting the water- $1H$ (and H-alpha?) T_2' decay analysis is solid, as I see theoretical problems, and hence I wonder if the τ_{rot} correlation times are trustworthy.

See our response to Point 5ii above. Slower molecular tumbling results in dipolar couplings being averaged less efficiently, just as it results in CSA being averaged less efficiently. Therefore, although dipolar couplings likely affect the apparent transverse relaxation rates, the qualitative trend that faster relaxation corresponds to slower molecular tumbling still holds.

In this context, the sentence “Transverse relaxation is driven by molecular tumbling, with slower relaxation indicative of shorter rotational correlation times (τ_{rot} .)” tells only part of the story, and I would not leave it written as it is. In fact, what is seen here is not necessarily relaxation, but an apparent coherence decay which comprises relaxation and most likely also coherent effects.

We agree with the reviewer that our signal decay is a mixture of relaxation and decay from coherent effects. Thus, we have changed the phrase “transverse relaxation” to “apparent transverse relaxation” and have added in the following sentence for clarification:

“Because water in our samples is not truly isotropic, coherent effects can accelerate the apparent transverse relaxation, such that it is faster than what is expected from molecular tumbling alone.”

6. There are open questions regarding the $1H R1\rho$ dispersion experiments, and the interpretation of the results. In fact, the evolution of the coherence under a spin-lock, at rather low MAS frequency (14 kHz) is fairly complex. First, contains evolution that is strictly independent of dynamics, i.e. in this context it is to be considered an “artefact”, in the sense that it would occur even if the molecules were static. Second, the $R1\rho$ decay contains parts due to dynamics, and here one can cite fluctuations of the chemical shift (which is the only thing the authors consider), but also fluctuations of the dipolar coupling and CSA. All these contributions to the observed $R1\rho$ decay – those that are dependent on dynamics and those that are not – depend on the RF field strength. Among the “coherent evolution” part (written under “First” above) I want to mention in particular the dipolar and CSA evolution – which the manuscript then nicely describes in the context of Figures 5 and S6. Under the present conditions, these

effects give rise to very fast coherence decay (plus oscillations) at RF fields somewhere from 7 to 30 kHz. This is precisely the range for which dispersions are reported in Figure 3d. Figure S6b actually shows that enhanced R1rho is expected for low RF fields – without any molecular motion or chemical shift fluctuations!

Among the “incoherent evolution” part (“Second” above), is indeed the expected Bloch-McConnell-type dispersion, but also relaxation effects arising due to dipolar/CSA couplings, and in particular when the RF field is close to the rotary resonance conditions. The latter is sometimes referred to as Nerrd effect. Thus, again in the range from 7 to 30 kHz, I would expect higher R1rho, while as the RF is further away from the rotary resonance conditions (i.e. at higher RF field), the R1rho should decrease. This is exactly what is observed. Therefore, I wonder how much (if any) of the relaxation dispersion can truly be ascribed to isotropic chemical-shift fluctuations, given that one can identify several effects that may produce very similar effects, some of which are even independent of dynamics. What is the argument to claim that all other effects than are negligible?

The NERRD effect should result in bumps centered around $0.5, 1,$ or $2 \Omega_r$. This is not what is observed. Instead we observe a Lorentzian centered at 0, which is the characteristic relaxation dispersion curve associated with isotropic chemical shift fluctuations. If our relaxation dispersion profile was from the NERRD effect we would expect relaxation to slow down at rf field strengths below Ω_r . See **Fig. 21** from Schanda & Ernst, *Prog. Nucl. Mag. Res. Spec.* **2016**, DOI: 10.1016/j.pnmrs.2016.02.001.

26

P. Schanda, M. Ernst/*Progress in Nuclear Magnetic Resonance Spectroscopy* 96 (2016) 1–46

Fig. 21. Conformational exchange as seen by ^{15}N $R_{1\rho}$ relaxation-dispersion experiments under MAS, obtained by stochastic Liouville simulations. An exchange between two states is assumed populated to 90% and 10% respectively, and differing in the bond orientation (CSA, dipole), and the ^{15}N isotropic chemical shift ($\Delta\nu = 300$ Hz). Different jump angles are simulated, as shown in the figure, and the exchange rate was 1000 s^{-1} . The MAS frequency was 40 kHz, and the B_0 field strength was 14.1 T. The red dashed area is shown in a zoom view on the right. Dashed curves show the case in which the isotropic chemical shift difference between the two states is zero, i.e. where only fluctuations of the anisotropic interactions occur.

I understand that one possible argument could be that the bulk water does not show dispersion.

But this is not a valid argument, because in the bulk water (freely tumbling), the dipolar/CSA interactions are clearly averaged to zero, which eliminates all the “coherent evolution part” above, as well as the “Nerrd” part above.

I also have problems understanding how the data from Figures 3d and Figure 5a go together. The data in these two experiments have been collected with the exact same pulse sequence, to my understanding. Figure 5 shows that at the rotary resonance conditions (14 kHz, 28 kHz) the signal decays very rapidly, and oscillates. This is expected for the presence of coherent dephasing (my “coherent evolution” part above). If these curves were fitted exponentially, I would estimate the rate constant to be ca. 1-2 ms, i.e. a rate constant of ca. 500 – 1000 s⁻¹. This is very far from the values reported in Figure 3d. How come? Is this just because the exact rotary resonance conditions were left out? I doubt, as the RF inhomogeneity will make it such that fast decay will occur even 1-2 kHz away (-> reported in Figure S6d).

Yes, the exact rotary resonance conditions were left out from **Figure 3d**. If the only “coherent” effect that is dominating signal decay is the ¹H CSA, then the matching conditions are actually very narrow (**Fig. S6b, right**). It is only ¹H homonuclear dipolar couplings that have relatively broad ranges in which the coherent effects result in the decay of transverse coherence. For CSA recoupling, being 1-2 kHz away from the recoupling condition results in barely any faster coherence decay. See our experimental optimization of the recoupling conditions below, which show that barely any coherent dephasing is observed when the rf power is >1 kHz off from the matching condition. Therefore, the narrow matching conditions observed in our recoupling experiments further support CSA, rather than ¹H homonuclear dipolar coupling, being the dominant coherent effect.

Let's assume for a moment that everything else than isotropic chemical-shift fluctuations can really be neglected. I wonder if the chemical-shift difference would make sense. If the exchange is really between the two water pools, as claimed, then I expect the chemical-shift difference to be very small (0.1-0.3 ppm), rather than the claimed >2 ppm. In other words, the dispersions are too large for what I think is a realistic chemical-shift difference. Thus, the pretty large dispersions point to something in addition to (or in place of) the isotropic chemical-shift fluctuations. So maybe my hypothesis that the dispersions contain a substantial part of coherent dephasing and/or Nerrd effects could hold true.

As mentioned above, if the relaxation dispersion were due to the NERRD effect or contributions from coherent dephasing, then we would expect the apparent relaxation to be slower at Ω_1 values below the rotary resonance recoupling conditions. However, we do not see this; the $T_{1\rho}$ relaxation rate decreases monotonically with increasing spin-lock field strengths (Fig. 3d), with the fastest relaxation occurring at the weakest applied spin-lock fields. Therefore, the relaxation dispersion in channel water is due to isotropic chemical shift differences. The only labile sidechain that could exchange with water at a rate faster than 1,000 s⁻¹ is histidine. However, we expect chemical change to be an order of magnitude faster at pH 7.5 than pH 4.5, while in our relaxation dispersion experiments we observe faster exchange at pH 4.5. Taken together, these data indicate that the relaxation dispersion arises from exchange of isotropic chemical shifts between two water populations. This isotropic shift difference between two water pools is indeed larger than we expected, but is not unheard of. Buried and hydration water interacting with

ubiquitin in a reverse micelle have a 1.1 ppm ^1H chemical shift difference (Nucci...Wand, *Nat. Struct. Mol. Biol.* **2011**, DOI: 10.1038/nsmb.1955), while water adsorbed to alumina and silica oxides has a chemical shift range of 2.6 ppm (Gun'ko...Turov, *Langmuir*. **1999**, DOI: 10.1021/la9809372). We have now added a more thorough explanation of our attribution of the relaxation dispersion to exchange between two water populations in the **Supporting Information**.

7. The recoupling experiments also have their own challenges. These experiments are very sensitive to the exact setting of the RF field, as illustrated in Figure S6d. The differences reported in the order parameters (Figure 5a) could easily be “obtained” by mis-setting the RF field strength by 1-2 kHz (I assume). Given that these two experiments have been done with different samples, and thus necessarily with different RF calibrations/tuning etc., it is not trivial to be sure that the RF field was properly set in each sample.

For each recoupling condition we meticulously optimized the proton rf power. To do this, we first measured the rf power level by finding a null in the ^1H - ^{13}C CP spectrum when the initial ^1H excitation pulse is set to a 180° pulse length for the given rf power. We then run the spin-lock recoupling experiment with a 1 ms spin-lock period and find at what ^1H rf power the aliphatic ^{13}C signal is minimized, which would indicate that the recoupling condition has been met (**Fig. S6d**). We first vary the ^1H rf power by 0.2 dB across a 2.0 dB range. We then check the signal intensity 0.1 dB above and below the minimum value from the first optimization step. After the recoupling experiments are run, we double check the rf matching condition using the same optimization routine to ensure that the power levels have not drifted during the course of the experiment.

Below we have plotted the 1D ^{13}C spectra from this optimization routine for the pH 4.5 sample under the $2\Omega_r$ ($n=2$) recoupling condition. The highest power used (-14.90 dB) is plotted in red and overlaid with subsequent conditions to show the differences in signal intensity. The integrated signal intensity is plotted as a function of the ^1H spin-lock rf power in the upper right. These plots have been added as an SI figure (**Fig. S7a**).

We note that a 0.1 dB difference in ^1H rf power would result in a 1.16% error in the rf field strength, which would correspond to a 0.32 kHz uncertainty at 28 kHz and 0.16 kHz uncertainty at 14 kHz. Therefore, we are sure that the experimental ^1H rf field strengths differ from the optimal matching condition by less than 0.16 kHz and 0.32 kHz at the Ω_r ($n=1$) and $2\Omega_r$ ($n=2$) matching conditions, respectively. We also note that a misset power level affects the minimum intensity of the S/S_0 plot more than it affects the timing that this minimum is reached and also does not significantly affect the short-time dephasing of the signal (**Fig. S6d**). Therefore, the fact that our dephasing curves reach approximately the same minimum intensity, but at different times and that the $2\Omega_r$ ($n=2$) matching condition has significantly different short-time dephasing for the low and high pH samples suggests that the water ^1H CSA is indeed greater for the low pH sample compared to the high pH sample.

Furthermore, when we lowered the temperature by 6 K, where the power levels and shim changed and needed to be reoptimized, we obtained essentially the same recoupling curves, which further supports that we did indeed find the true optimal recoupling condition through our optimization routines. See **Figure S7b** below.

Furthermore, in theory the CSA parameters obtained from the $n=1$ and $n=2$ rotary resonance conditions should be identical, but they are not: in the low-pH sample one finds 2.0 ppm and 1.6 ppm for the two experiments, and for the high-pH sample it is 1.6 ppm and 1.2 ppm. The mismatch between the CSA parameters obtained for a given sample is as large as the difference between the sample. The former is not discussed, but the latter is interpreted in terms of molecular mechanisms.

The $n=1$ condition also has contributions from homonuclear dipolar couplings. We have now found a previous published example of spin-lock recoupling at the $n = 2$ rotary resonance condition to measure ^1H CSA that we were previously unfamiliar with (Duma...Bodenhausen, *Chem. Commun.*, **2008**: DOI: 10.1039/B801154K). In this paper, the authors used the same experiment as our ^1H -detected spin-lock recoupling pulse sequence (now **Fig. S3b**) to measure the CSA of hydroxyl protons and water protons in octasilicate, a layered hydrous sodium silicate. The authors also observed significantly greater deviations in the $n = 1$ condition, which they attribute to contributions from ^1H - ^1H homonuclear dipolar couplings and thus rely on the $n = 2$ rotary resonance condition for determining the magnitude of the CSA. They observed a water ^1H CSA of 5.0 ppm, which is larger than what we observed in the current study, as would be expected for the greater degree of order within hydration water in octasilicate. We have now added citations to this paper and added the following sentence:

“Although CSA is the dominant coherent effect for water, contributions from ^1H - ^1H homonuclear dipolar couplings can result in enhanced dephasing at the ω_r recoupling condition⁴⁸; therefore, we report CSA parameters extracted from the $2\omega_r$ -matching conditions to compare the relative water anisotropy between our two samples.”

This paper confirms our use of these experiments to measure ^1H CSA and further explains why we rely on the $n = 2$ rotary resonance condition for determining the water CSA.

I was also curious why the data of the spin-lock experiments presented in Figure 5a and 5b are not identical. I assume that the data shown in Figure 5b are from a repeat experiment, but

otherwise identical to those shown in Figure 5a. I would claim that the data are not very well reproducible. What would be the fitted CSA parameters from the data shown in Figure 5b?

We have now added plots in **Fig. S6e** to show the RMSD between our recoupling experiments and simulations for different CSA values. We note that the $n = 1$ condition has an RMSD that is approximately 3-fold higher than the $n = 2$ condition, further supporting that there are nuclear interactions unaccounted for in the $n = 1$ simulation.

In **Fig. 5b** we used the same mixing times for the spin-lock experiment as for the RECCR experiments. We cannot sample the RECCR experiments as finely as the traditional spin-lock experiments, and thus reran those experiments using the RECCR timings. The power levels for the spin-lock recoupling experiments with the RECCR spin-lock time points were not optimized immediately prior to running these experiments, leading to a slight missetting of the RF power. We have replaced the data in **Fig. 5b** with the data from **Fig. 5a**. Below are the RMSD plots for the spin-lock recoupling experiments with the fine sampling along with using the RECCR timing. We note that although the matching conditions were not as optimal for the experiments with the RECCR timing, the best fit CSA parameters differ by only 0.1 ppm and our conclusion that the pH 4.5 channel has greater anisotropy holds.

8. The paper reports that in the low-pH state (open state) water reorients more rapidly. This is based on the reorientational correlation time that comes from $1H T_2'$ measurements (Table S1), with all its theoretical problems (see my point 5). At the same time, the oscillations seen in $1H$ spin-lock experiments (pulse sequence: Figure S2b; data: Figure 5a middle and right panels). I

find it difficult to see how water 1H can have slower T2' relaxation (which the authors translate to faster reorientation using BPP theory) and at the same time have higher order parameters. To me this sounds like a contradiction. In the Discussion section, the authors address this as follows:

“The surprising finding of faster water dynamics and higher water anisotropy can be rationalized by the fact that water needs to reorient to break and form hydrogen bonds with adjacent water molecules to permit Grotthuss hopping.”

But I do not see how Grotthuss hopping enters the picture here. Grotthuss hopping means that a rearrangement of hydrogen bonding/covalent bonding looks like an effective translation of molecules, although in reality it is not, in the sense that a given atom is actually not moving. NMR is not sensitive to translational motion, only to rotations and fluctuations of inter-atomic distances. Hence, NMR can only see actual movement of motions, but not a “pseudomovement” as in the Grotthuss mechanism.

We have reworded the sentence in question to

“The surprising finding of faster water dynamics and higher water anisotropy can be rationalized by the fact that water molecules that reorient quickly break and form hydrogen bonds with adjacent water molecules to permit proton exchange and thus Grotthuss hopping”.

We are not claiming that our data directly indicate that the Grotthuss mechanism is occurring; we are claiming that faster rotational motion combined with a preferred orientation can be rationalized to permit Grotthuss hopping through water-wire like pathways in which water molecules are properly oriented to exchange protons and are rapidly making and breaking hydrogen bonds.

Thus, for me the apparent contradiction of shorter T2' (Fig 3b) and lower order parameter (Fig 5a) in the high-pH state is not resolved. As pointed out in 5. and 7. above, I see several possibilities why the experiments and their interpretation may go wrong, but I don't see how to reconcile the data. Could the authors please explain their view?

The order parameter obtained from fitting our spin-lock recoupling experiments is not related to the rate of motion, but rather related to the time averaged ensemble orientation of water within the channel. Therefore, a higher order parameter reflects water molecules being more likely to be in particular orientations, but says nothing about how quickly they are reorienting. We attribute the higher order parameter in the low pH channel to the negative dipole of water molecules preferentially aligning to point towards the positively charged H19 tetrad. We were surprised to find that the water orientation is more anisotropic for the low pH channel, given its faster rotational motion, but these two observations are not in contradiction to each other.

We lowered the temperature by 6 K to investigate whether dynamics or chemical exchange affect our observed CSA order parameter. At this lower temperature we obtain nearly identical recoupling curves (as mentioned above). Therefore, the recoupled order parameter is indeed not strongly affected by the dynamics of water molecules in the channel. We thus conclude that it is

most strongly affected by the charge state of the H19 tetrad. We have added these recoupling curves at 267 K to **Fig. S7b** and added the following statement to the main text:

“¹³C-detected ¹H spin-lock recoupling experiments conducted at lower sample temperatures produced identical recoupling curves (Fig. S7b), which experimentally supports that it is the charge state of histidine that controls water orientation anisotropy since chemical exchange and rotational motion are both slowed down at lower temperatures.”

By the way, isn't the Grothuss mechanism impacted by pH? Thus, does changing the pH by 3 units (1000-fold more H+) impact the argument that the authors make here?

Decreasing the pH by 3 units increases hydronium ion concentration and thus a greater likelihood of transporting protons by the Grothuss mechanism. This, combined with our findings that water in the pH 4.5 channel is more dynamic and has a greater average orientation preference, supports that the Grothuss mechanism would be more efficient in the low pH, open channel than in the high pH, closed channel.

Minor comments:

Page 1, bottom: “our recently determined 1.5 Å resolution structures of BM2 in lipid bilayers”. The term relating to a certain resolution (here: 1.5 Å) is so closely linked to crystallography, in the minds of almost all structural biologists, that most readers will very likely assume that these are crystal structures. But in fact, they are ssNMR structures. I think this should be made clear.

And the bundle RMSD in an NMR structure calculation is not quite the same as the resolution that crystallographers use. I think some clarification would help.

We agree, and have revised the sentence to:

“A direct comparison of water properties between the open and closed channels is made possible by our recently determined solid-state NMR structures of BM2 in lipid bilayers whose structural ensemble of 10 lowest-energy structures has an RMSD of 1.5 Å²⁷. ”

Typo in the SI text, first page: “approximatiely”

We thank the reviewer for catching this typo and have corrected it in the SI.

The axis in Figure 2d is a bit odd. On the left (and inside the panel) are residue names. However, I see only 8 residue names, but 9 (black) data points; the data point are not really aligned with the ticks next to the residue names. Not clear who is who in this plot.

The 8 residue names refer to key residues to orient the reader as to where the channel coordinate axis lies along the M2 channel. The 9 data points are taken from MD results that use the channel axis coordinates on the right side of the figure. The residue names are placed on the figure to orient the reader and show what the channel axis coordinates refer to. The data points do not directly correspond to particular residues. We have updated the figure caption to reflect this and now explicitly label the vertical axis on both the right and left side of the panel. The relationship between channel coordinate axis position and average residue position can be found in **Table S4**.

The caption of Figure S6 states: “Experiments began with x-magnetization and the amount of magnetization remaining on spin #1 was monitored as a function of spin-lock time.” The reference to spin #1 is not useful, because the spins are not numbered in the figure. Are the four water molecules symmetry related, i.e. they all have the same environment?

We have now numbered the spins in **Figure S6**; the four water molecules are indeed symmetry related, though the two protons on each water molecule are in different magnetic environments. Ultimately, this does not matter for CSA, but would matter for homonuclear dipolar couplings in which the protons are coupled to each other with through space interactions.

Figure S6d: What do the simulations look like at RF fields above 28 kHz? Are they symmetrical, i.e. is the behavior at 29 kHz identical to the one at 27 kHz? Figure S6 (simulations of recoupling curves): what is the rationale for the chosen RF field distribution? This is not a realistic distribution for an NMR probe, in my view. It would better be approximated by a Gaussian distribution of RF fields, whereby the tail on the lower RF field side is longer than on the larger RF-field side. Arguably the details of this distribution change very little for the outcome of the fit.

RF fields above 28 kHz are nearly symmetric to RF fields below 28 kHz. We have added the following sentence to our Methods section:

“We note that an rf inhomogeneity of 110%-100% results in approximately the same dephasing curve as an rf inhomogeneity of 90%-100%, e.g. dephasing is roughly symmetric about the matching condition.”

The small differences are not enough to affect our fit. Below is a figure in which we compare what the simulated recoupling curve would look like for RF distributions ranging from 25.2 to 28 kHz and 30.8 to 28 kHz.

We chose the RF field distribution given in **Figure S6** based upon the fact that we expect the rf fields inside our coil to be quite homogeneous and for rf fields at the measured value of 14 or 28 kHz to contribute significantly more than rf fields that are not at the measured value. We empirically tried several different weighting functions (including $y=1$, $y=x$, $y=x^2$, $y=x^4$) and found the x^4 weighting function to produce dephasing curves that best recapitulate our experimental data.

We ran a long nutation experiment and took its Fourier transform to determine the inhomogeneity. We used an rf field strength of 35 kHz to avoid any issues with the rotary resonance condition and assume the inhomogeneity at 35 kHz is not all that different from the inhomogeneity at 14 or 28 kHz. The general profile of our experimental rf inhomogeneity is similar to the assumed profile used in this study, however our nutation experiment suggests that the rf field is more inhomogeneous than what we used to fit our experiments. We note that for the Ω_r matching condition, the inhomogeneity profile measured from our nutation experiment would reach a minimum value of ~ 0.2 at about 1 ms. However, experimentally we observe this matching condition to reach a minimum of about 0 at 1 ms. Therefore, we believe that the nutation experiment does not accurately reflect the true rf homogeneity for our probe. It is possible that this issue arises from the non-negligible rise and fall times of the coil for short pulses, which would become negligible for the length of cw irradiation used in our experiments and therefore have decided to stick with the rf inhomogeneity profile initially used in this study.

Figure S6b: the simulations are shown for the (unrealistic) case that there is no RF inhomogeneity. For illustration purposes that's ok, but it should be stated explicitly that in a realistic scenario the conditions are broader.

As the reviewer says, this was for illustration purposes. We have revised the caption to:

“We note that the simulations in (b) and (c) depict the unrealistic condition of having no rf inhomogeneity, however they serve as an illustrative map of what rotary resonance matching conditions result in coherent recoupling for dipolar couplings and CSA.”

REVIEWERS' COMMENTS:

Reviewer #1 (Remarks to the Author):

The revisions carried out by the authors and the new simulations performed as well as the expansion of the simulation method section address my concerns. The manuscript greatly improved, providing the reader the information to understand better the background and the computational tools used to study water molecules in the influenza B M2 protein channel, and the scientist the details of the simulations in case a similar work is going to be repeated.

The work done by the authors to correct and edit the paper, add new data, and address my comments is remarkable and I now believe that the manuscript is in a good shape to be published.

Reviewer #2 (Remarks to the Author):

The authors have provided a very thorough analysis of the issues that I have raised. I thank them for the clarifications and corrections.

From the NMR perspective, I think that this paper provides now sufficient details to allow the reader to draw conclusions. Some of the findings remain surprising to me (e.g. the >2 ppm chemical shift difference from the relaxation-dispersion experiment, discussed now in the Supp Info). However, the arguments are plausible and future studies will tell if this value makes sense.

I found from the comments of the other reviewer that the computational part seemed to have serious shortcomings. I cannot judge those with certainty.

I am in favor of accepting the manuscript, if the computational part is robust.